# BDNF/TrkB signaling endosomes in axons coordinate CREB/mTOR activation and protein synthesis in the cell body to induce dendritic growth in cortical neurons

**Guillermo Moya-Alvarado[1†], Reynaldo Tiburcio-Felix[2], María Raquel Ibáñez[2], Alejandro A Aguirre-Soto[2], Miguel V Guerra[1,2], Chengbiao Wu[3], William C Mobley[3], Eran Perlson[4], Francisca C Bronfman[2]\***

[1]Department of Physiology, Faculty of Biological Sciences and Center for Aging and Regeneration), Pontificia Universidad Católica de Chile. Av. Libertador Bernardo O´Higgins, Santiago, Chile; [2]NeuroSignaling Lab (NESLab), Center for Aging and Regeneration (CARE-UC), Institute of Biomedical Sciences (ICB), Faculty of Medicine, and Faculty of Life Sciences, Universidad Andrés Bello, Santiago, Chile; [3]Department of Neurosciences, University of California, San Diego, San Diego, United States; [4]Department of Physiology and Pharmacology, Sackler Faculty of Medicine; Sagol School of Neuroscience, Tel Aviv University, Tel Aviv, Israel

**\*For correspondence:** francisca.bronfman@unab.cl

**Present address:** [†]Department of Biology, Johns Hopkins University, Baltimore, United States

**Competing interest:** The authors declare that no competing interests exist.

**Abstract** Brain-derived neurotrophic factor (BDNF) and its receptors tropomyosin kinase receptor B (TrkB) and the p75 neurotrophin receptor (p75) are the primary regulators of dendritic growth in the CNS. After being bound by BDNF, TrkB and p75 are endocytosed into endosomes and continue signaling within the cell soma, dendrites, and axons. We studied the functional role of BDNF axonal signaling in cortical neurons derived from different transgenic mice using compartmentalized cultures in microfluidic devices. We found that axonal BDNF increased dendritic growth from the neuronal cell body in a cAMP response element-binding protein (CREB)-dependent manner. These effects were dependent on axonal TrkB but not p75 activity. Dynein-dependent BDNF-TrkB-containing endosome transport was required for long-distance induction of dendritic growth. Axonal signaling endosomes increased CREB and mTOR kinase activity in the cell body, and this increase in the activity of both proteins was required for general protein translation and the expression of Arc, a plasticity-associated gene, indicating a role for BDNF-TrkB axonal signaling endosomes in coordinating the transcription and translation of genes whose products contribute to learning and memory regulation.

## Editor's evaluation

The results in this study represent an advance over previous experiments since long distance signaling is demonstrated in brain neurons, whereas earlier work was concerned with peripheral neurons. Also, this article is the first to show in a conclusive manner that intracellular signaling in brain neurons results in increased transcription and protein production. The novelty is derived from long-distance signaling that is proposed for the development of circuits in the brain.

## Introduction

Somatodendritic growth during CNS development is essential for the establishment of proper connections (*Jan and Jan, 2010*). In addition, the morphology of dendritic arbors is maintained and undergoes plastic changes through a process that requires activity-dependent transcription factors (TFs), protein translation, and extracellular neurotrophic factors (*Sutton and Schuman, 2006*; *Wong and Ghosh, 2002*; *Yap et al., 2018*).

Brain-derived neurotrophic factor (BDNF) is a well-known neurotrophin whose transcription is regulated by neuronal activity and is required for the maintenance of dendrites of adult neurons in the CNS. BDNF binds two receptors, tropomyosin kinase receptor B (TrkB) and the p75 neurotrophin receptor (p75). The classical trophic, growth-promoting actions of BDNF largely involve activation of TrkB, as mutant mice with inducible deletion of TrkB receptors exhibit a significant reduction in the number of dendritic arbors and degeneration of cortical neurons (*Xu et al., 2000*; *Horch and Katz, 2002*; *Cheung et al., 2007*; *Yan et al., 1997*; *Huang and Reichardt, 2003*).

Dendritic arborization mediated by TrkB activation in neurons is triggered by signaling pathways such as the ERK1/2 and the cAMP response element-binding protein (CREB) pathways (*Finkbeiner et al., 1997*; *Xing et al., 1998*; *Andres-Alonso et al., 2019*). CREB also mediates activity-dependent transcription by regulating early gene expression (*Flavell and Greenberg, 2008*; *Yap and Greenberg, 2018*). In addition, activation of the class 1 phosphoinositide 3-kinase (PI3K)/Akt/mammalian target of rapamycin (mTOR) pathway increases the translation of a subset of mRNAs, including those that encode proteins related to BDNF-mediated regulation of dendritic size and shape in the CNS (*Takei et al., 2004*; *Ravindran et al., 2019*; *Schratt et al., 2004*; *Leal et al., 2014*; *Kumar et al., 2005*; *Minichiello et al., 2002*; *Dijkhuizen and Ghosh, 2005*).

A large amount of evidence has shown that a complex of neurotrophins and neurotrophin receptors continue signaling inside the cell in specialized organelles named signaling endosomes, where neurotrophin receptors encounter specific signaling adaptors (*Grimes et al., 1996*; *Lazo et al., 2014*; *Debaisieux et al., 2016*). Several studies have shown that endocytosis of TrkB is required for survival, dendritic arborization, and cell migration (*Zheng et al., 2008*; *Zhou et al., 2007*), indicating that endosomal signaling has a role in the physiological responses to TrkB. Consistent with these observations, studies from our laboratory have shown that there is a functional relationship between BDNF/TrkB signaling and the early recycling pathway in the regulation of dendritic morphology and CREB-dependent transcription of plasticity genes (*Lazo et al., 2013*; *Moya-Alvarado et al., 2018*; *González-Gutiérrez et al., 2020*).

Long-distance communication between neurotrophins in the target tissue and cell bodies has been described in the PNS and requires microtubule-associated molecular motors, such as cytoplasmic dynein (*Lazo et al., 2014*; *Reck-Peterson et al., 2018*). During this process, NGF/TrkA signaling endosomes originating from the synaptic terminal propagate distal neurotrophin signals to the nucleus to allow transcriptional regulation, inducing neuronal survival (*Cosker and Segal, 2014*; *Scott-Solomon and Kuruvilla, 2018*; *Riccio et al., 1997*). Although it has been shown that BDNF is retrogradely transported along axons of cortical neurons, whether central neurons respond to distal neurotrophic signals is a relevant and understudied question in the neurobiology field. Indeed, no studies have directly assessed whether axonal BDNF signaling endosomes induce long-distance effects (*Zhao et al., 2014*; *Olenick et al., 2019*; *Zhou et al., 2012*; *Stuardo et al., 2020*; *Cohen et al., 2011*). Here, we studied whether axonal BDNF/TrkB signaling has a dendritic growth-promoting effect in the cell bodies of cortical neurons. We demonstrated for the first time that dynein-mediated transport of BDNF/TrkB signaling endosomes can modulate neuronal morphology. We also explored the BDNF/TrkB downstream signaling mechanism that regulates this process and showed that axonal BDNF-TrkB signaling endosomes coordinate the phosphorylation of CREB and PI3K-mTOR pathway-related proteins in cell bodies to increase protein translation and dendritic branching.

## Results

### Axonal BDNF signaling promotes dendritic branching in a TrkB-dependent manner

Using cortical neurons in compartmentalized cultures in microfluidic chambers, we evaluated whether axonal stimulation with BDNF promotes dendritic arborization in a TrkB activity-dependent manner.

We used two different systems: rat cortical neurons incubated with the tyrosine kinase inhibitor K252a (*Tapley et al., 1992*) and cortical neurons derived from TrkB[F616A] knock-in mice. This mutation in TrkB enables pharmacological control of TrkB kinase activity using small molecule inhibitors, including 1-naphthylmethyl 4-amino-1-tert-butyl-3-(p-methyl phenyl) pyrazolo[3,4-d] pyrimidine (1NM-PP1), which selectively, rapidly, and reversibly inhibits TrkB activity. In the absence of such inhibitors, TrkB is fully functional (*Chen et al., 2005*). We also used cortical neurons derived from p75 exon III knockout mice (p75 KO[exonIII]) (*Lee et al., 1992*) to assess the dependence of dendritic growth and morphology on p75 expression and activity.

To evaluate morphological changes in neurons, we transfected cortical neurons in compartmentalized cultures (DIV 6) with plasmids expressing EGFP and then performed MAP2 immunostaining after fixing the neurons (*Figure 1A*). To identify the neurons that projected their axons to the axonal compartment (AC) and to assess leakage of media from the AC into the cell body compartment (CB), we used fluorescently labeled cholera toxin B subunit (Ctb). Ctb is internalized and retrogradely transported along neuronal axons and accumulates in the Golgi apparatus in the cell body (*Escudero et al., 2019*; *Figure 1A* and *Figure 1—figure supplement 1*). Chambers with neurons that showed Ctb accumulation in the cell body but not those in which Ctb diffusely labeled the soma of neurons were analyzed (*Figure 1—figure supplement 1*). Only neurons in which the cell body was labeled with Ctb were used for quantification. Since BDNF is also released from dendrites (*Matsuda et al., 2009*), axonal BDNF was selectively stimulated by incubating somas in the CB compartment with TrkB-Fc chimera (*Shelton et al., 1995*). TrkB-Fc neutralized the activity of endogenous BDNF released by neurons; when TrkB-Fc was not added to the CB compartment, axonal BDNF induced a nonsignificant increase in the number of distal dendrites (*Figure 1B*).

The experiment presented in *Figure 1B* and explained above revealed that the addition of BDNF to axons led to increases in the numbers of primary dendrites, branching points and overall dendritic arbors in the cell bodies of cortical neurons from rats (*Figure 1C–F*) and mice (*Figure 1G–J*). Furthermore, we found that this effect required axonal TrkB activation, as treating the axons of rat cortical neurons with K252A or treating cortical neurons from TrkB[F616A] knock-in mice with 1NM-PP1 abolished the effect of axonal BDNF on neuronal morphology (*Figure 1C–J*).

To evaluate the contribution of p75 to the dendritic arborization induced by axonal BDNF, we used cortical neurons from p75 KO[exonIII] mice. Using the same experimental design described in *Figure 1A and B*, we prepared cultures derived from the cross of p75 KO[exonIII] heterozygous mice. The littermates were wild type (p75WT), heterozygous (p75HET), and homozygous (p75KO) mice.

We observed that the number of branching points and primary dendrites induced by axonal stimulation with BDNF was unchanged by the p75 genotype (*Figure 2A–C*). Nevertheless, Sholl analysis of dendritic morphology revealed that the basal morphology of the dendritic arbors was altered in the absence of p75. While p75KO neurons possessed a smaller number of primary dendrites, they had an increased number of distal dendrites (*Figure 2D*); however, this phenotype was rescued by axonal BDNF (*Figure 2E*).

Altogether, we provided direct evidence for BDNF long-distance signaling contributing to dendritic branching in neurons from the CNS. Also, we showed that this effect depends mainly on the activity of TrkB (and not p75) in axons.

## CREB activity is required for long-distance BDNF signaling

Several groups have shown that axonal BDNF signaling promotes nuclear CREB phosphorylation in different neuronal models (*Zhou et al., 2012*; *Watson et al., 1999*; *Deinhardt et al., 2006*; *González-Gutiérrez et al., 2020*); however, the physiological relevance of CREB activation in this process has not been evaluated. To evaluate CREB activation, we used a well-characterized polyclonal antibody against CREB phosphorylated at S133 (*Lonze and Ginty, 2002*; *González-Gutiérrez et al., 2020*) and studied the time course of CREB phosphorylation induced by axonal BDNF treatment. At DIV 5, we incubated the AC with Ctb-555 overnight to identify neurons with axons in the AC (*Figure 3A*). BDNF induced an increase in nuclear CREB phosphorylation as soon as 30 min, with nuclear CREB phosphorylation peaking at two time points: at 30 min and 180 min (*Figure 3A and B*).

To confirm that CREB activity is required for dendritic arborization, we stimulated the axons of cortical neurons whose cell bodies had been incubated with KG501 for 48 hr with BDNF (*Figure 3C*). KG501 is a small molecule that disrupts the interaction between CREB and CREB-binding protein

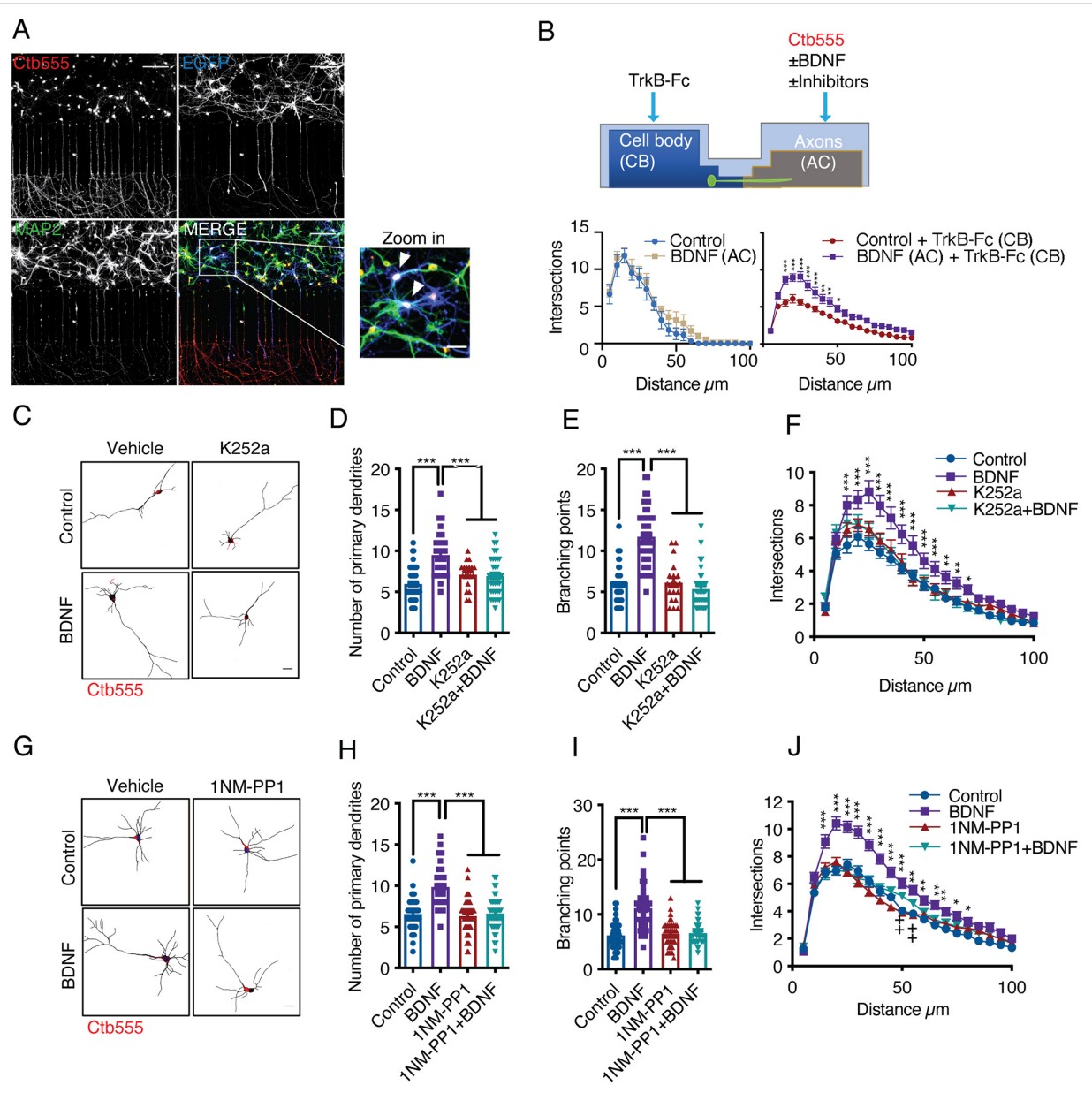

**Figure 1.** Addition of BDNF to axons promotes dendritic branching in a TrkB-dependent manner. (**A**) Representative images of compartmentalized microfluidic chambers (400 µm long and 15 µm wide microgrooves) after addition of BDNF/Ctb-555. Ctb-555 (upper left panel), EGFP (upper right panel), and MAP2 (lower left panel) were visualized by confocal microscopy (Scale bar = 50 µm). The lower right panel is a merged image of the three signals. The white arrows in the zoomed-in panel indicate neurons costained with EGFP/MAP2/Ctb-555 (Scale bar = 20 µm), that is, neurons that project axons through the microgrooves. (**B**) Upper panel, experimental design used to study retrograde BDNF signaling. DIV six neurons were transfected with a plasmid expressing EGFP. TrkB-Fc (100 ng/mL) was applied to the CB. BDNF (50 ng/mL) in addition to fluorescently labeled Ctb-555 was added to the AC in the presence or absence of different inhibitors. BDNF and TrkB-Fc were administered for 48 hr to evaluate dendritic arborization. Finally, the neurons were fixed, and immunofluorescence for MAP2 was performed. Lower panels, graphs showing Sholl analysis of dendritic arborization of compartmentalized rat cortical neurons stained with Ctb-555. Left graph, BDNF (50 ng/mL) was applied or not to the AC for 48 hr (n=10–15 neurons, from two independent experiment). Right graph, TrkB-Fc (100 ng/mL) was added to the CB, and BDNF (50 ng/mL) was applied or not to the AC for 48 hr (n=37 neurons, from two independent experiment). The results are expressed as the mean ± SEM. ***p<0.001. The Sholl analysis data were analyzed by two-way ANOVA followed by Bonferroni's multiple comparisons post-hoc test. (**C**) Representative images of the CB (red indicates Ctb-555) of compartmentalized rat cortical neurons whose axons were treated with DMSO (control), K252a (200 nM), BDNF (50 ng/mL) or BDNF following preincubation with K252a (**D–F**). Quantification of primary dendrites (**D**) and branching points (**E**) and Sholl analysis (**F**) for neurons labeled with EGFP/MAP2/Ctb-555 under each experimental condition. n=40–45 neurons from three independent experiments. (**G**) Representative images of the CB (red indicates Ctb-555) of compartmentalized mouse TrkB[F616A] cortical neurons whose axons were treated with DMSO (control), 1NM-PP1 (1 µM), BDNF

*Figure 1 continued on next page*

*Figure 1 continued*

(50 ng/mL), or BDNF following preincubation with 1NM-PP1. (**H–J**) Quantification of primary dendrites (**H**) and branching points (**I**) and Sholl analysis (**J**) for neurons labeled with EGFP/MAP2/Ctb-555 under the four different experimental conditions described in G. Scale bar = 20 µm. n=40–45 neurons from three independent experiments. *p<0.05, **p<0.01, ***p<0.001,++p < 0.01, the 1NM-PP1 group vs. the 1NM-PP1+BDNF group. The data were analyzed by one-way ANOVA followed by Bonferroni's multiple comparisons post-hoc test (**D, E, H and I**). Two-way ANOVA followed by Bonferroni's multiple comparisons post-hoc test (**F and J**). The results are expressed as the mean ± SEM.

The online version of this article includes the following source data and figure supplement(s) for figure 1:

**Source data 1.** Addition of BDNF to axons promotes dendritic branching in a TrkB-dependent manner.

**Figure supplement 1.** Experimental design used to study retrograde BDNF signaling in compartmentalized cortical neurons.

(CBP) (*Best et al., 2004*). We found that KG501 abolished the increase in dendritic arborization induced by treating axons with BDNF (*Figure 3D–F*). We also evaluated whether p75 is required for the phosphorylation of CREB upon BDNF stimulation. Consistent with the results presented in *Figure 2*, the absence of p75 did not impact BDNF-induced CREB phosphorylation (*Figure 3—figure supplement 1*), indicating that in our model, p75 was not required for BDNF-induced CREB phosphorylation. Since CREB has been widely described as a significant regulator of neurotrophin-induced survival of PNS neurons (*Finkbeiner et al., 1997*; *Bonni et al., 1999*; *Lonze and Ginty, 2002*), we assessed whether KG501 has an effect on neuronal survival under our experimental conditions by assessing neuronal apoptosis using TUNEL staining. We found that after 48 hr of treatment, KG501 did not affect neuronal survival. We used oligomycin A, an inhibitor of oxidative phosphorylation, as a positive control to induce apoptotic death of cortical neurons (*Figure 3—figure supplement 2*).

To validate the findings of *Figure 3D–F* with a molecular biology approach, we cotransduced cortical neurons (DIV 4) with adeno-associated virus (AVV) serotype 1 (AAV1) expressing dominant negative CREB (CREB-DN-EGFP) or control (EGFP) together with AAV1 expressing mCherry. The expression was for four days in the presence or absence of axonal BDNF and TrkB-Fc was present in all conditions in the cell body compartment (*Figure 3G*). About 75% of neurons were cotransduced with EGFP or CREB-DN-EGFP and mCherry. We found that the expression of CREB-DN-EGFP did not reduce the number of mCherry-positive cells containing a Hoechst-labelled nucleus compared to the control (*Figure 3H*). This result indicates that CREB-DN-EGFP did not cause neuronal cell death and is consistent with the results presented in *Figure 3—figure supplement 2*, where inhibition of CREB by KG501 did not increase neuronal apoptosis.

On the other hand, the fluorescence associated with mCherry clearly labeled neurites in the cell body (CB) compartment of cortical neurons grown in compartmentalized cultures (*Figure 3G1*). The addition of BDNF in axons increased mCherry-labeled neurites in the CB compartment of EGFP-expressing cells but not in cultures expressing CREB-DN-EGFP (*Figure 3I and J*). The fact that the total amount of neurites is diminished in neurons expressing CREB-DN-EGFP compared to EGFP (non-significant differences) expressing cells is consistent with the idea that neurons require CREB activity to maintain normal morphology. Similar results are obtained when adult pyramidal neurons (cortical layers II-III) in the sensory-motor cortex express CREB-DN-EGFP in vivo (*Figure 3—figure supplement 3*). This circuit requires BDNF and TrkB to sustain normal neuronal morphology (*Andreska et al., 2020*; *Shimada et al., 1998*). When transduced with AAV1 expressing mCherry, the cortico-callosal pathway is labeled (*Figure 3—figure supplement 3A*). Furthermore, when these neurons were cotransduced with AAV1 expressing CREB-DN-EGFP/mCherry for three weeks, the cell body size, the number of neurons, the apical dendrite diameter and branching were reduced compared to neurons expressing EGFP/mCherry (*Figure 3—figure supplement 3*). Apoptotic cell death, as indicated by fragmentation of the nucleus revealed by Hoechst staining, was not observed (*Figure 3—figure supplement 3B*), indicating that the expression of CREB-DN-EGFP did not induce apoptotic cell death, consistent with the in vitro results presented in *Figure 3*.

Altogether, we demonstrated that CREB activation is required for long-distance induction of dendritic branching by BDNF in compartmentalized cortical neurons in vitro. In addition, CREB was also needed for sustaining normal neuronal morphology in long-projecting neurons, such as the pyramidal neurons of the sensory-motor cortex.

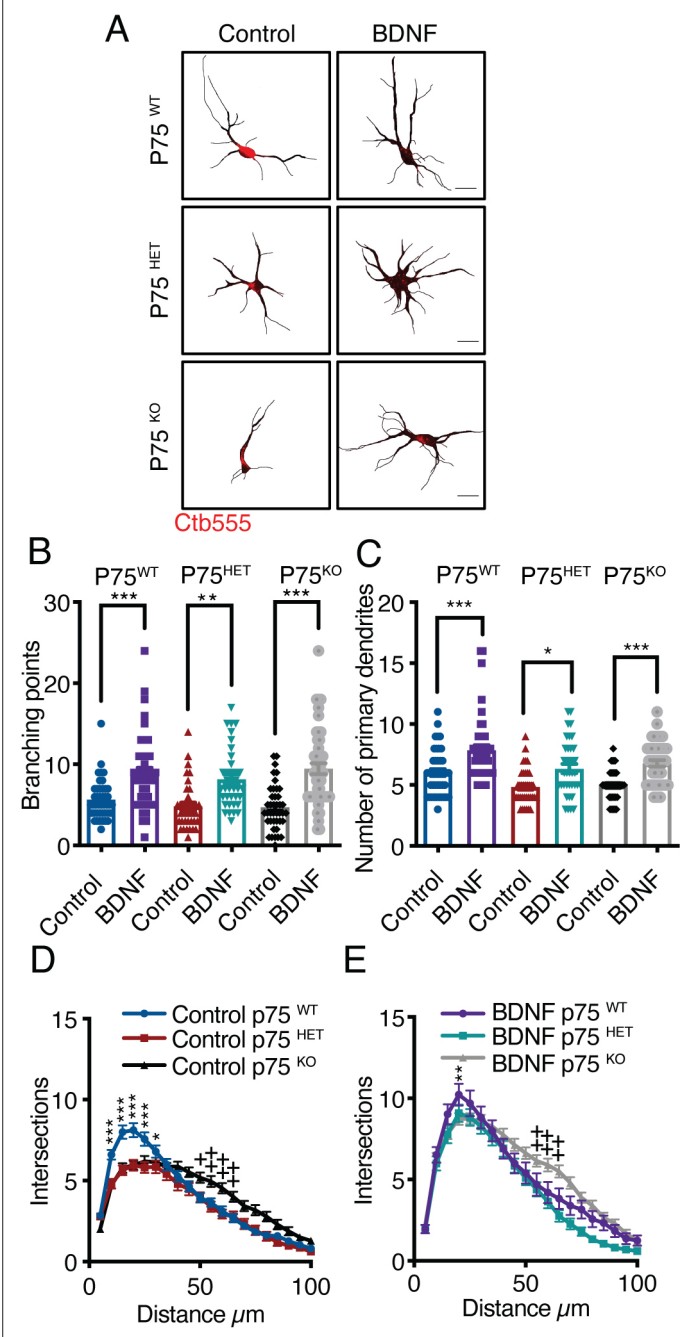

**Figure 2.** Axonal BDNF increases dendritic arborization in a p75-independent manner. (**A**) Representative images of the CB (red indicates Ctb-555) of compartmentalized p75WT, p75HET, or p75KO mouse cortical neurons whose axons were treated with BDNF (50 ng/mL). Quantification of branching points (**B**) and primary dendrites (**C**) in MAP2/Ctb-555-labeled neurons from mice of the different genotypes described in A. (**D**) Sholl analysis of cultured neurons from mice of each genotype following application of TrkB-Fc to the CB (nonstimulated). (**E**) Sholl analysis of cultured neurons from mice of each genotype following addition of TrkB-Fc to the CB and BDNF to the AC. n=40–50 neurons from two independent experiments (3 mice per experiment). Scale bar = 10 µm *p<0.05, **p<0.01, ***p<0.001,++p < 0.01, neurons from p75KO mice vs. neurons from p75WT mice. The data were analyzed by one-way ANOVA followed by Bonferroni's multiple comparisons post-hoc test (**B and C**). Two-way ANOVA followed by Bonferroni's multiple comparisons post-hoc test (**D and E**). The results are expressed as the mean ± SEM.

The online version of this article includes the following source data for figure 2:

**Source data 1.** Axonal BDNF increases dendritic arborization in a p75-independent manner.

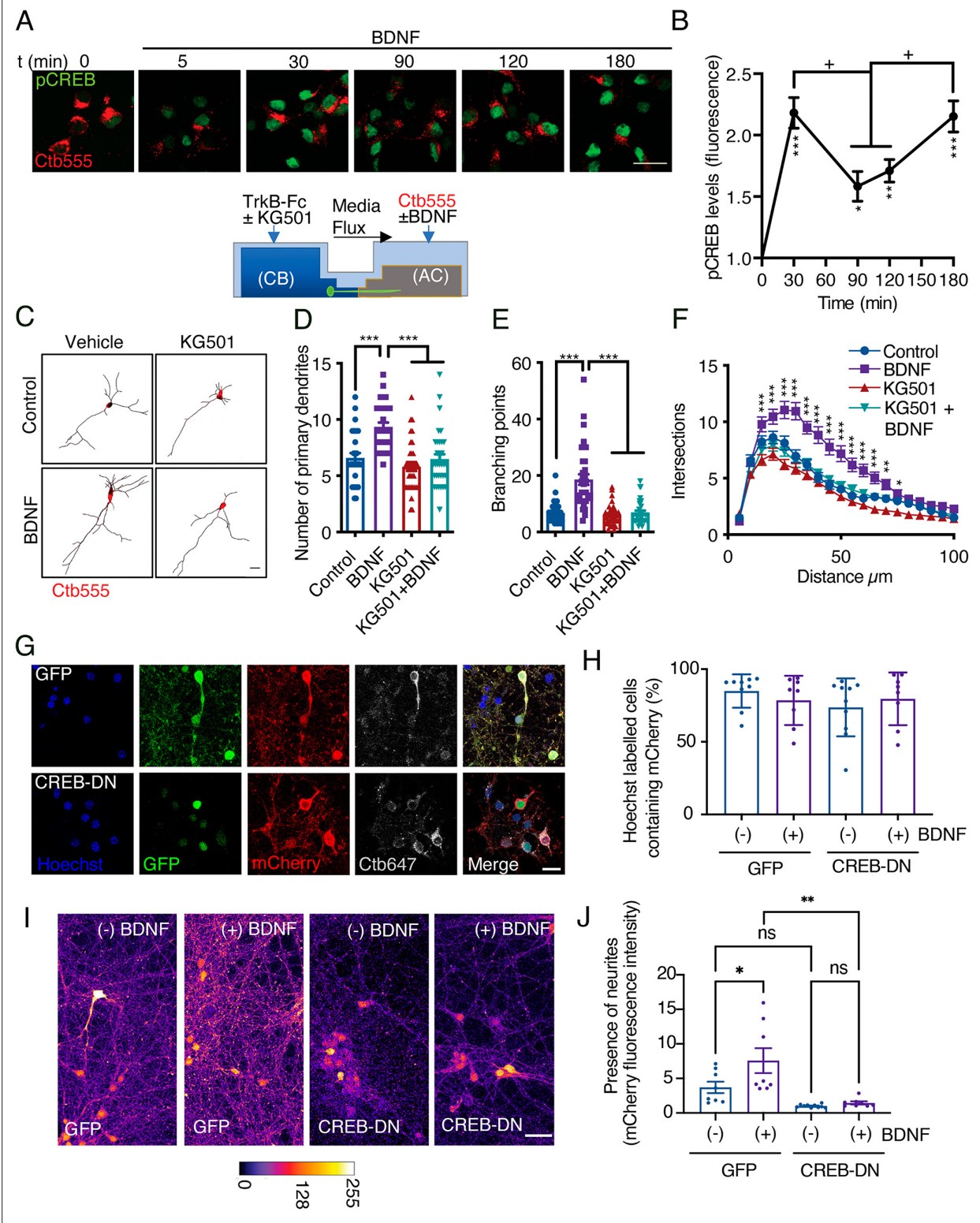

**Figure 3.** CREB activity is required for long-distance axonal BDNF-induced dendritic arborization. (**A**) Lower panel, schematic representation of the protocol used to evaluate CREB phosphorylation induced by axonal BDNF and the requirement of CREB for dendritic arborization induced by BDNF using the drug KG501(panels **C, D, E and F**). Upper panel, representative figures of nuclear activated CREB (pCREB, S133) immunostaining in control neurons and neurons whose axons were stimulated with BDNF for different times. Ctb-555 was added to the AC of compartmentalized neurons overnight, and after 90 min of serum deprivation, BDNF (50 ng/mL) was added to the AC for 5, 30, 90, 120, or 180 min. pCREB is shown in green and Ctb-555 is shown in red. Scale bar = 20 μm. (**B**) Quantification (arbitrary units, A.U.) of pCREB fluorescence in the nucleus in the presence and absence

*Figure 3 continued on next page*

*Figure 3 continued*

of BDNF. Only neurons positive for Ctb-555 were included in the analysis. n=30–40 neurons from two independent experiments. *p<0.05, **p<0.01, ***p<0.01 vs. the control group;+p < 0.05 vs. the 90 min and 120 min BDNF treatment groups. The results are expressed as the mean ± SEM. The data were analyzed by one-way ANOVA followed by Bonferroni's multiple comparisons post-hoc test. (**C**) Representative images of the CB (red indicates Ctb-555) of compartmentalized neurons whose cell bodies were treated with DMSO (control), KG501, BDNF, or BDNF in the presence of KG501. TrkB-Fc (100 ng/mL) and KG501 (10 μM) were applied to the CB for 1 hr Then, BDNF (50 ng/mL) was added to the AC for 48 hr in the presence of Ctb-555, and the CB was treated with or without KG501. Finally, the neurons were fixed, and immunofluorescence for MAP2 was performed. Scale bar = 20 μm. (**D–F**) Quantification of primary dendrites (**D**) and branching points (**E**) and Sholl analysis (**F**) for neurons labeled with EGFP/MAP2/Ctb-555 under each experimental condition. n=30–35 neurons from three independent experiments. *p<0.05, **p<0.01, ***p<0.001. The data were analyzed by one-way ANOVA followed by Bonferroni's multiple comparisons post-hoc test (**D and E**). Two-way ANOVA followed by Bonferroni's multiple comparisons post-hoc test was used to analyze the Sholl analysis data (**F**). The results are expressed as the mean ± SEM. (**G**) Representative images of compartmentalized cortical neurons transduced at DIV 4 with AAV1 expressing EGFP/mCherry (upper figures) or CREB-DN-EGFP/mCherry (lower figures). At DIV 7, neurons were treated with Ctb-647 in the AC. After 16–24 hr, neurons were fixed and mounted on Mowiol containing Hoechst. The level of cotransduced cells was about 75%. Scale bar = 20 μm. (**H**) Quantification of the percentage of cells containing Hoechst that also contained mCherry in the different treatments. EGFP(GFP)/mCherry or CREB-DN-EGFP(CREB-DN)/mCherry transduced neurons were treated (+) or not (-) with BDNF for 4 days since DIV 4 in the AC. Then, the percentage of cells containing Hoechst that also contained mCherry was calculated. Non-significant different were found between groups. The data were analyzed by two-way ANOVA followed by Turkey's multiple comparisons test. (**I**) Representative images of compartmentalized cortical neurons treated as explained in G and H. The fluorescence associated with mCherry is observed using the fire LUT from ImageJ. Scale bar = 20 μm. (**J**) Quantification of mCherry-associated fluorescence in neurites standardized by the mean value of the group of chambers transduced with CREB-DN/mCherry and non-treated with BDNF (-). n=8–9 chambers from three independent experiments. *p<0.05, **p<0.01. The data were analyzed by two-way ANOVA followed by Turkey's multiple comparisons test. The results are expressed as the mean ± SEM.

The online version of this article includes the following source data and figure supplement(s) for figure 3:

**Source data 1.** CREB activity is required for long-distance axonal BDNF-induced dendritic arborization.

**Figure supplement 1.** Evaluation of CREB activation in different p75 strains.

**Figure supplement 1—source data 1.** Evaluation of CREB activation in different p75 strains.

**Figure supplement 2.** Evaluation of neuronal survival after KG501 treatment.

**Figure supplement 2—source data 1.** Evaluation of neuronal survival after KG501 treatment.

**Figure supplement 3.** Expression of a dominant-negative mutant of CREB (CREB-DN-EGFP) in the pyramidal neurons of the sensory-motor cortex of mouse brain induced dendritic retraction and shrinkage of neuronal body size.

**Figure supplement 3—source data 1.** Expression of a dominant-negative mutant of CREB (CREB-DN-EGFP) in the pyramidal neurons of the sensory-motor cortex of mouse brain induced dendritic retraction and shrinkage of neuronal body size.

## Dynein-dependent transport of endocytosed TrkB in axons is required for long-distance BDNF signaling

In axons, long-range retrograde transport of organelles and signaling molecules is achieved by the minus-end microtubule molecular motor cytoplasmic dynein (*Hirokawa et al., 2010*). Several groups have shown that in CNS neurons, BDNF and TrkB are efficiently retrogradely transported (*Zhou et al., 2012*; *Olenick et al., 2019*). To further confirm that this process is dynein dependent, we monobiotinylated BDNF in vitro and labeled it with streptavidin quantum dots (QDs) for in vivo imaging (*Stuardo et al., 2020*). We found that BDNF-QDs was efficiently targeted to the retrograde transport pathway and that the retrograde transport of BDNF-QDs was reduced by approximately 80% by inhibition of dynein with ciliobrevin D (CilioD), a specific inhibitor of dynein ATP activity (*Sainath and Gallo, 2015*; *Figure 4A and B*). To evaluate whether axonal dynein inhibition reduces long-distance BDNF signaling, we evaluated whether axonal BDNF-induced nuclear phosphorylation of CREB and dendritic arborization in axons in the presence and absence of CilioD (*Figure 4C*). We found that CilioD reduced CREB phosphorylation (*Figure 4D–F*) and dendritic arborization induced by the application of BDNF to axons (*Figure 4G–J*), which is consistent with the concept that CREB-dependent arborization induced by axonal BDNF is dependent on dynein transport of BDNF signaling endosomes.

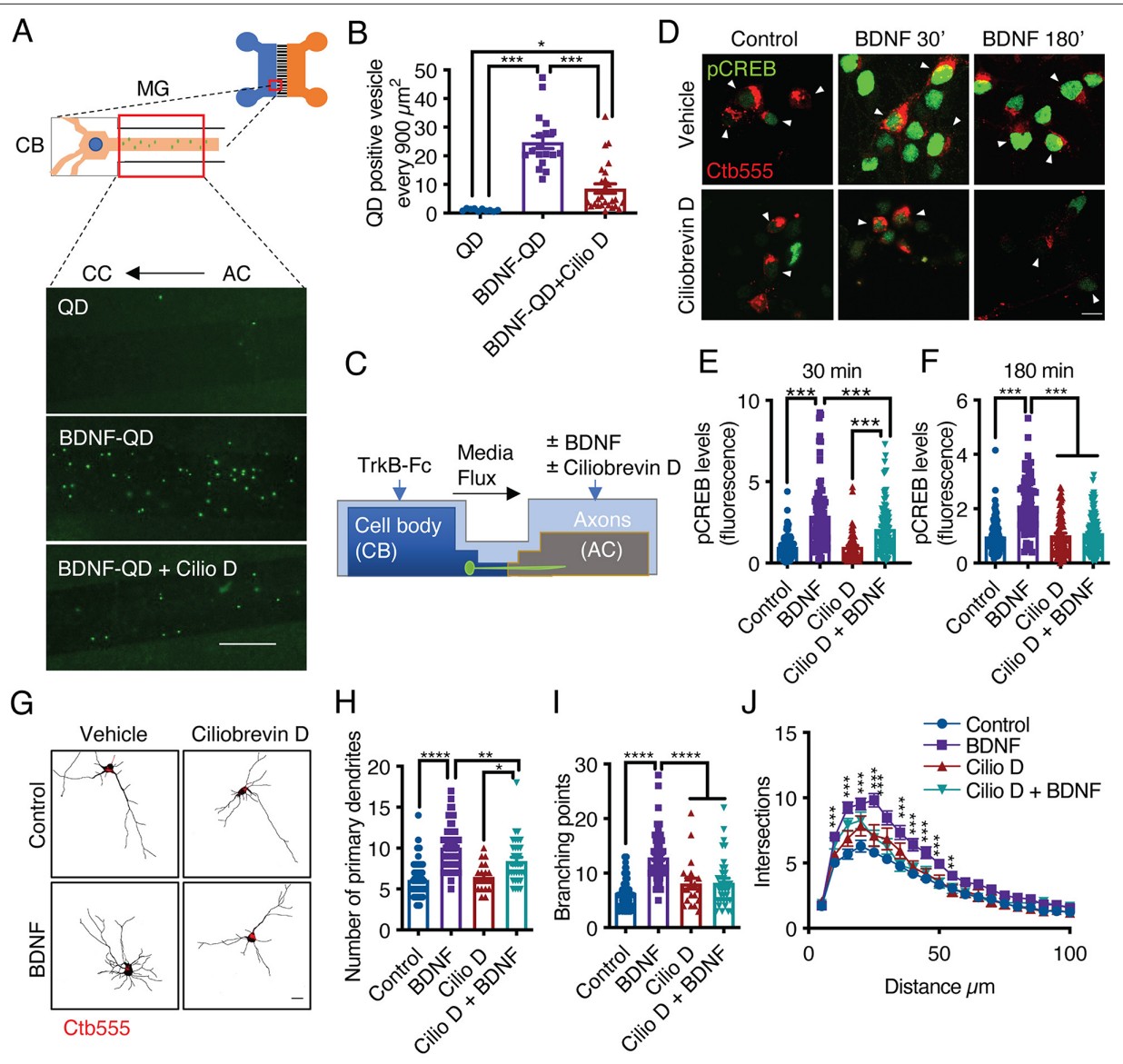

**Figure 4.** Dynein activity is required for phosphorylation of CREB and dendritic arborization induced by BDNF. (**A**) Schematic representation of the protocol used to evaluate the retrograde transport of BDNF. Biotinylated BDNF conjugated to streptavidin QD-655 (2 nM) was added to AC for 4 hr in the presence or absence of CilioD (20 μM). Then, after washing, the neurons were fixed, and the accumulation of BDNF-QDs in the proximal compartment of the microgrooves (MG) was evaluated. Unconjugated QDs were used as controls. Representative images of BDNF-QDs (green) in the proximal region of the MG under each condition. Scale bar = 10 μm (**B**) Quantification of QD-positive vesicles in every 900 μm$^2$ area of the proximal region of MG under each condition. N=36 MG from three independent experiments were analyzed. (**C**) Experimental design used to study dynein-dependent axonal BDNF signaling in compartmentalized cortical neurons. DIV five cortical neurons were retrogradely labeled with Ctb-555 (red) overnight. The next day, the culture medium was replaced with serum-free medium for 90 min, and CilioD was applied to the AC. Then, BDNF (50 ng/mL) was added to the AC for 30 or 180 min in the presence or absence of CilioD. Finally, the cultures were fixed, and pCREB (S133) was visualized by immunofluorescence. (**D**) Representative images of nuclear pCREB in neurons. Scale bar = 10 μm. (**E–F**) Quantification of (arbitrary units, A.U.) pCREB immunostaining in the nuclei of neurons labeled with Ctb-555 and stimulated with BDNF for 30 min (**E**) or 180 min (**F**). n=83–111 neurons from three independent experiments. (**G**) DIV six cortical neurons were transfected with EGFP, TrkB-Fc (100 ng/mL) was applied to the CB, and Ctb-555 and BDNF (50 ng/mL) with or without CilioD (20 μM) were added to the AC for 48 hr. Finally, the neurons were fixed, and immunofluorescence for MAP2 was performed. Representative images of the morphology of compartmentalized neurons that were labeled with Ctb-555 (red) and whose axons were treated with DMSO (control), CilioD, BDNF, or BDNF following pretreatment with CilioD for 48 hr. Scale bar = 20 μm. (**H–J**) Quantification of primary dendrites (**H**) and branching points (**I**) and Sholl analysis (**J**) for neurons labeled with EGFP/MAP2/Ctb-555 under each experimental condition described in G. n=34–65 neurons from three independent experiments. *p<0.05, **p<0.01, ***p<0.001, ****p<0.0001. The data were analyzed by one-way ANOVA followed by Bonferroni's multiple comparisons post-hoc test (**E, F, H, and I**). The Sholl analysis data were analyzed by two-way ANOVA followed by Bonferroni's multiple comparisons post-hoc test. The results are expressed as the mean ± SEM.

*Figure 4 continued on next page*

*Figure 4 continued*

The online version of this article includes the following source data for figure 4:

**Source data 1.** Dynein activity is required for phosphorylation of CREB and dendritic arborization induced by BDNF.

## The somatodendritic activity of signaling endosomes containing BDNF and active TrkB is required for long-distance signaling induced by BDNF in axons

Several reports have shown that endocytosed axonal BDNF is retrogradely transported to and accumulates in cell bodies (*Olenick et al., 2019*; *Zhou et al., 2012*; *Zhou et al., 2012*). These findings, together with the observations presented in this research, suggest that BDNF is transported along with TrkB via signaling endosomes to the cell body, where the BDNF-TrkB complex continues to signal.

To our knowledge, there is no evidence that active TrkB in axons, reaches the cell body and continues to signal to induce CREB activation. Thus, we design an experiment where we ought to activate TrkB in TrkB^F616A knock-in mouse neurons in compartmentalized cultures by applying BDNF to the AC and inhibited the activity of TrkB upon its arrival in the cell body by applying 1NM-PP1 to the CB. To achieve this goal, we first studied whether 1NM-PP1 can reduce TrkB activity after BDNF has activated the receptor in noncompartmentalized cultures. We stimulated neurons with BDNF for 30 min and then incubated them with 1NM-PP1 or DMSO for an additional 30 min. As a control, we preincubated the neurons with 1NM-PP1 for 1 hr to prevent the activation of TrkB (*Figure 5—figure supplement 1A*). As expected, preincubation of neurons with 1NM-PP1 prevented the activation of TrkB and Akt induced by BDNF (*Figure 5—figure supplement 1B*). In addition, we observed that treatment with 1NM-PP1 after BDNF stimulation decreased the phosphorylation of TrkB and Akt to levels like those before incubation (*Figure 5—figure supplement 1B–D*). Consistently, immunofluorescence revealed that 1NM-PP1 treatment after BDNF stimulation reduced the nuclear phosphorylation of CREB and the phosphorylation of 4E-BP1 and S6 ribosomal protein (S6r), two proteins downstream of mTOR activation induced by BDNF (*Figure 5—figure supplement 1E–H*). These results indicate that 1NM-PP1 can turn off TrkB after it is activated by BDNF.

Then, to evaluate whether axonal-derived activated TrkB reaches the cell body and is required for the propagation of axonal BDNF signaling, we performed transfection, pulse chase and NM-PP1 inhibition experiments as follows. We first transfected neurons with a plasmid expressing TrkB containing a Flag NH2-terminal epitope (Flag-TrkB) (*Lazo et al., 2013*) and then incubated the axons with a Flag-specific antibody at 4 °C, washed them, and treated them with BDNF at 37 °C (*Figure 5A*). After fixation, we evaluated the colocalization of Flag with pTrkB in microgrooves in proximity to the AC (distal microgrooves), in the vicinity of the CB (proximal microgrooves) and in the CB (*Figure 5B*). We found that signaling endosomes containing activated TrkB move along axons and reach the cell body upon stimulation of axons with BDNF. Subsequently, we directly assessed whether we could reduce TrkB activity in the cell bodies of TrkB^F616A mouse-derived neurons after stimulation of axons with BDNF. Somatodendritic TrkB activity was evaluated after applying 1NM-PP1 to the CB or AC (*Figure 5C*). Application of BDNF to the axons increased the immunostaining intensity of endogenous pTrkB in the CB (*Figure 5D and E*). Consistent with the observations in noncompartmentalized neuronal cultures, the application of 1NM-PP1 to the AC completely abolished the accumulation of pTrkB in the cell body (*Figure 5D and E*). Furthermore, the application of 1NM-PP1 to the cell body decreased the amount of activated TrkB in the cell body after stimulation of axons with BDNF (*Figure 5D and E*).

These results suggest that 1NM-PP1 inhibited the phosphorylation of endosomal TrkB after it arrived in the neuronal cell body. To further confirm this phenomenon, we studied the phosphorylation of CREB in the cell bodies of TrkB^F616A mouse neurons in compartmentalized cultures after stimulation of axons with BDNF in the presence or absence of 1NM-PP1. We found that somatodendritic TrkB activity was required for CREB phosphorylation induced by stimulation of axons with BDNF (*Figure 5F and G*). Suggesting that axonal endosomes containing BDNF and active TrkB reach the cell body, aiding to propagate axonal BDNF signaling to the nucleus.

We incubated fluorescently labeled BDNF (f-BDNF) and Ctb in the axons to further study this possibility for 6 hr. We found that both labels accumulated in the cell bodies of cortical neurons. Some vesicles contained both marks and others had one or the other (*Figure 6A*). Then, we performed experiments to find colocalization of f-BDNF vesicles with active TrkB in cell bodies using a commercial

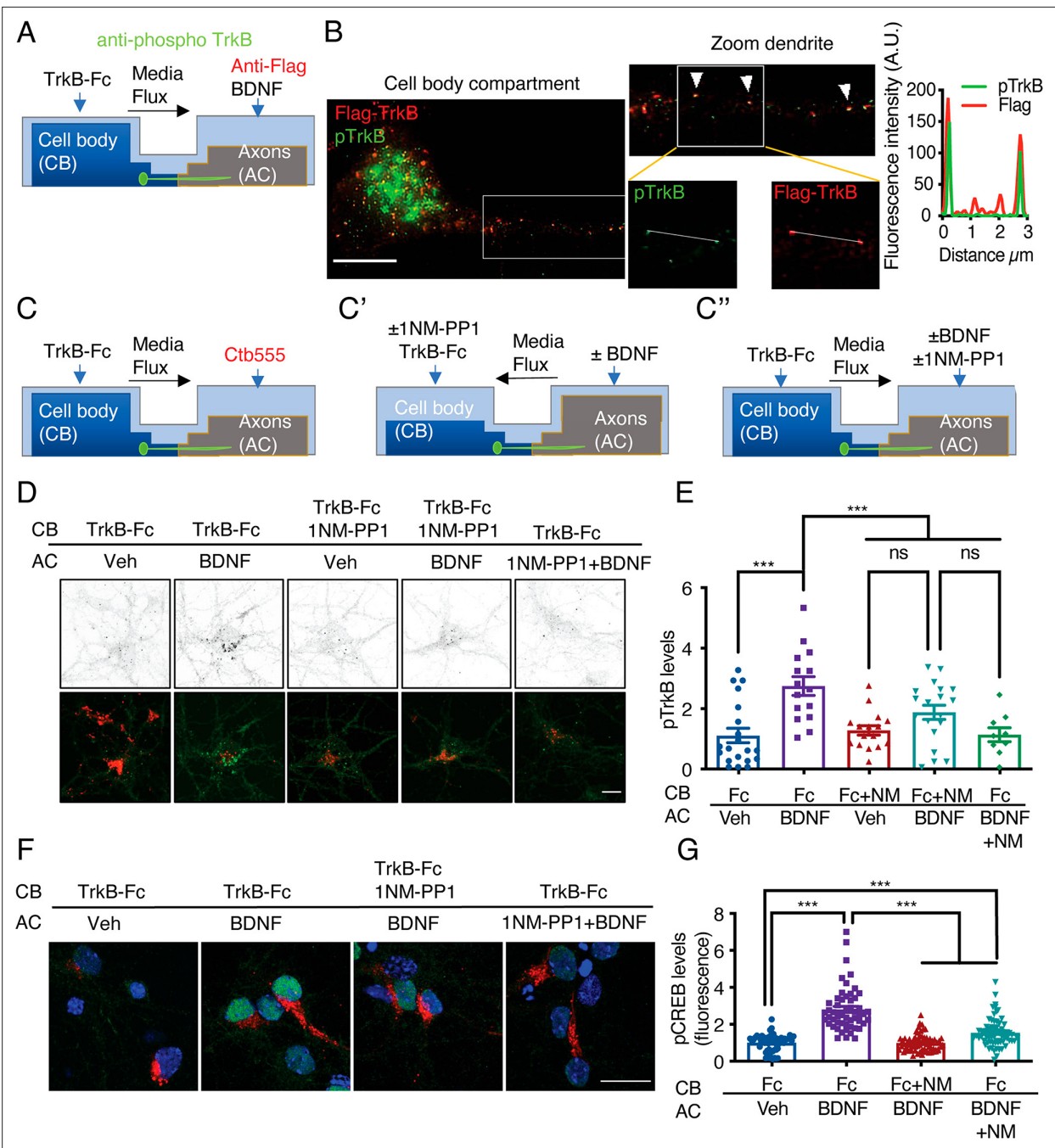

**Figure 5.** Somatodendritic activity of axonal TrkB is required for long-distance BDNF signaling. (**A**) Schematic representation of the protocol used to label axonal TrkB. DIV five compartmentalized cortical neurons were transfected with Flag-TrkB for 48 hr. At DIV seven, anti-Flag antibody was added to AC for 40 min at 4 °C. Then, neurons were incubated with or without BDNF (50 ng/mL) at 37 °C. Finally, immunofluorescence for pTrkB (Y816, pTrkB) (green) and anti-Flag (red) was performed. (**B**) Immunofluorescence of pTrkB and internalized Flag-TrkB in compartmentalized neurons whose axons were treated with BDNF, as shown in A. Right panel, representative images of neurons whose axons were treated with BDNF. Left panel, a magnified image of a proximal neuronal dendrite. The arrows indicate colocalization of Flag (red) and pTrkB (green). Scale bar, 5 μm. Right panel, graphs showing the fluorescence intensity pixel by pixel along the white lines shown in the right panels in B. The green line indicates pTrkB fluorescence, and the red lines indicate Flag-TrkB fluorescence. (**C-C"**) Schematic representation of the protocol used to stimulate neurons in D-G. (**C**) At DIV 6, Ctb-555 was added to the AC of cortical neurons from TrkBF616A mice overnight. The next day (DIV 7), the neurons were subjected to two treatments: (**C'**) depletion of B27 supplement for 1 hr in the presence of 1NM-PP1 (1 μM) in the CB, with flux toward the CB. Then, TrkB-Fc (100 ng/mL) and 1NM-PP1 were added to the CB, and BDNF (50 ng/mL) was applied or not to the AC for 3 hr. (**C"**) Neurons were depleted of B27 supplement for 1 hr in the presence or absence of 1NM-PP1 (1 μM) in the AC, with flux toward the AC. Then, TrkB-Fc was applied to the CB, and BDNF was added or not to the AC in the presence or absence of 1NM-PP1 for 3 hr. Finally, the neurons were fixed, and immunofluorescence for pTrkB (Y816) or pCREB (S133) was performed.

*Figure 5 continued on next page*

*Figure 5 continued*

(**D**) Representative images of pTrkB in the CB compartment of Ctb-positive neurons treated as described in A. Scale bar = 5 μm. Quantification of pTrkB levels in the cell body in each treatment group described in B. n=3 independent experiments. (**F**) Representative image of pCREB immunostaining in cortical neurons whose axons were stimulated with BDNF for 3 hr in the presence or absence of 1NM-PP1 in the CB (1NM-PP1/BDNF) or AC (1NM-PP1+1NM-PP1). Scale bar, 10 μm. (**G**) Quantification of pCREB levels in Ctb-positive neurons under each condition. n=78–86 neurons from three independent experiments. \*\*p<0.01, \*\*\*p<0.001. vs. the control group. The data were analyzed by one-way ANOVA followed by Bonferroni's multiple comparisons post-hoc test. The results are expressed as the mean ± SEM.

The online version of this article includes the following source data and figure supplement(s) for figure 5:

**Source data 1.** Somatodendritic activity of axonal TrkB is required for long-distance BDNF signaling.

**Figure supplement 1.** 1NM-PP1 can reduce TrkB activation in TrkBF616A mouse cortical neurons after BDNF treatment.

**Figure supplement 1—source data 1.** 1NM-PP1 can reduce TrkB activation in TrkBF616A mouse cortical neurons after BDNF treatment.

**Figure supplement 1—source data 2.** 1NM-PP1 can reduce TrkB activation in TrkBF616A mouse cortical neurons after BDNF treatment.

phospho- antibody against phospho-tyrosines 706/707 (Y706/707) located in the catalytic domain of the intracellular portion of TrkB (*Tacke et al., 2022*). We found consistent colocalization of f-BDNF with pTrkB in vesicles larger than 10 μm$^2$ in the cell body (*Figure 6C, D and H*) and MAP2-positive dendrites (*Figure 6E, F and H*). These results show that upon the addition of BDNF to axons, BDNF is retrogradely transported to cell bodies together with its cognate active receptor TrkB.

Together, our findings are the first to demonstrate that axonal BDNF and endogenous active TrkB are in signaling endosomes in cell bodies where their activity is required to propagate axonal BDNF signals to the nuclei of CNS neurons.

## The somatodendritic activity of CREB and PI3K-mTOR is required for protein translation and dendritic branching induced by axonal BDNF

BDNF-mediated dendritic arborization is achieved via activation of the PI3K/Akt/mTOR signaling pathway, which increases protein translation in the cell body and dendrites (*Dijkhuizen and Ghosh, 2005*; *Kumar et al., 2005*). Additionally, PI3K has been shown to mediate retrograde NGF signaling in sympathetic neurons (*Kuruvilla et al., 2000*). Furthermore, it was reported that axonal endosomes are positive for TrkB and Akt in hippocampal neurons (*Goto-Silva et al., 2019*), suggesting that axonal PI3K/Akt signaling may contribute to long-distance dendritic arborization induced by BDNF. Thus, we applied BDNF to the AC of compartmentalized cortical neurons in the presence or absence of LY294002, a potent general PI3K inhibitor (*Figure 7A*). Interestingly, the application of LY294002 to axons did not affect dendritic arborization induced by axonal BDNF (*Figure 7B–E*). This result was not due to a lack of LY294002 effect in cortical cultures since, confirming previous results (*Finster-wald et al., 2010*), LY294002 reduced BDNF-induced dendritic arborization in noncompartmentalized cortical cultures (*Figure 7—figure supplement 1*). Furthermore, inhibition of PI3K activity in axons did not reduce the transport of BDNF-QDs (*Figure 7F and G*). This result indicates that PI3K activity is not required for the retrograde transport of axonal BDNF signals. Therefore, we investigated whether PI3K signaling in the somatodendritic compartment is required for axonal BDNF-induced dendritic arborization by applying LY294002 to the CB of compartmentalized cortical cultures (*Figure 7H*). We observed that the application of LY294002 to the CB inhibited dendritic arborization induced by axonal BDNF (*Figure 7I–L*), suggesting that PI3K activation in the cell body is required for long-distance signaling induced by axonal BDNF.

Once PI3K is activated, it activates the mTOR kinase pathway by reducing the GAP activity of the Rheb GAP, leading to the activation of mTOR (*Garza-Lombó and Gonsebatt, 2016*). First, to evaluate BDNF-mediated activation of the mTOR pathway in cortical neurons, we stimulated noncompartmentalized cultures with BDNF for 1 hr in the presence or absence of LY294002 (a PI3K inhibitor) and Torin1, a specific inhibitor of mTOR that reduces the activity of both the TORC1 and TORC2 complexes (*Liu et al., 2010*). Western blotting revealed that BDNF promoted the phosphorylation of TrkB, Akt, S6r, and 4E-BP1 in cortical neurons and that LY294002 and Torin1 fully inhibited the phosphorylation of Akt, S6r and 4E-BP1 but not TrkB (*Figure 7—figure supplement 2*), as expected. Then, we evaluated whether the mTOR pathway is involved in increasing protein translation upon arrival of pTrkB signaling endosomes in the cell body. To this end, we evaluated the time course of 4E-BP1 phosphorylation in the neuronal cell body induced by axonal BDNF

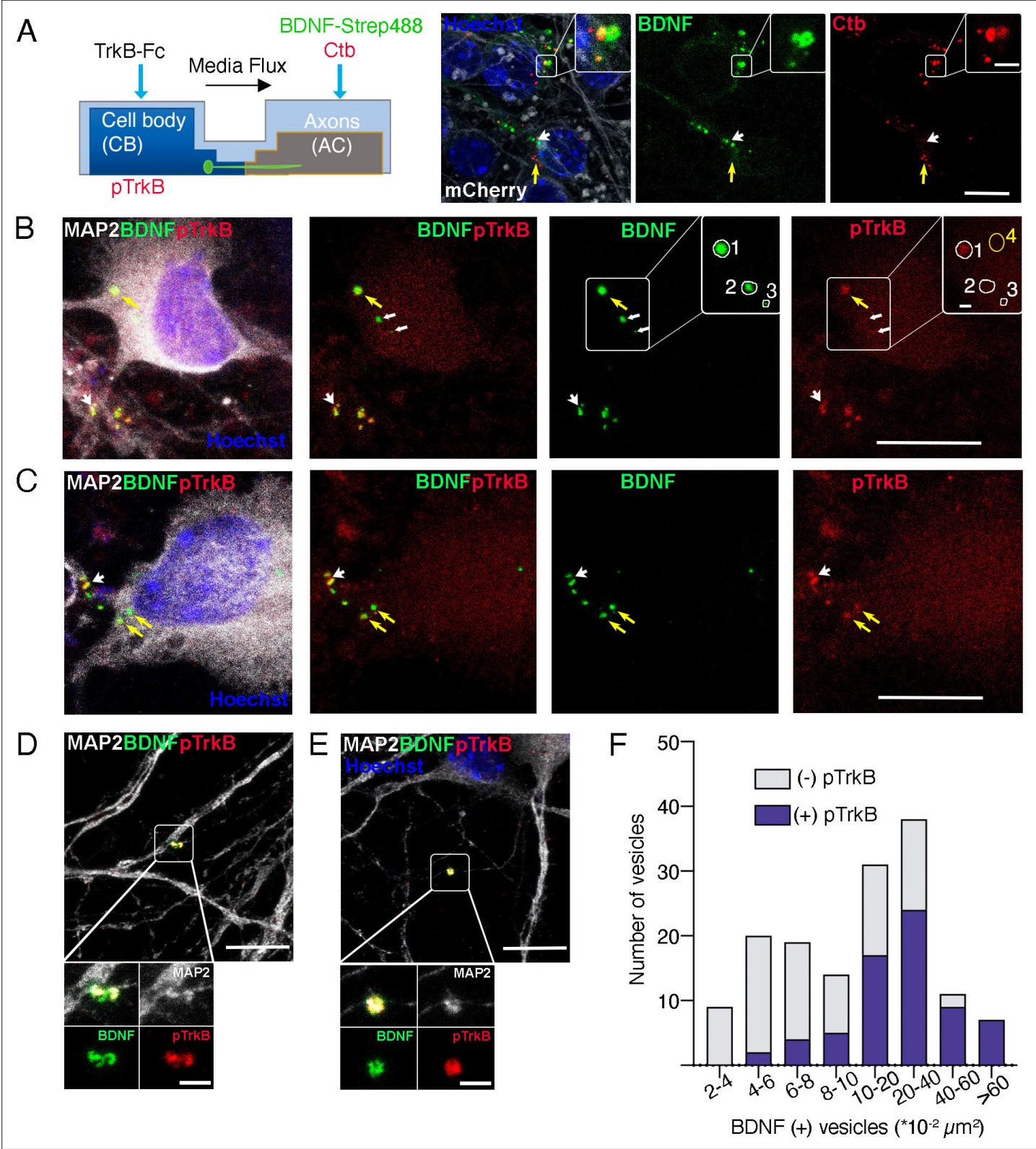

**Figure 6.** Axonal BDNF colocalizes with active TrkB in signaling endosomes within cell bodies and dendrites of compartmentalized cortical neurons. (**A**) Left panel, schematic representation of the protocol used to stimulate neurons. Right panel, DIV 4 neurons were transduced with AAV1 expressing mCherry. After 4 days, Ctb-647 (Ctb, red) and biotinylated BDNF conjugated to streptavidin Dylight 488 (BDNF-strep488, green) were added to the axonal compartment (150 ng/mL) for 6 hr. Then, cultures were fixed, mounted in Mowiol containing Hoechst, and prepared for visualization.

*Figure 6 continued on next page*

*Figure 6 continued*

Representative images are shown where yellow arrows indicate BDNF-strep488 only vesicles, and white arrows indicate Ctb only vesicles. Scale bar = 10µm. Inset, magnification of endocytic structures where BDNF-strep488 and Ctb are partially colocalizing (observed in yellow Scale bar = 2 µm). (**B–E**) DIV 8 neurons were serum-deprived for 90 min and BDNF-strep488 was added to the axonal compartment (150 ng/mL) for 6 hr. Then, neurons were fixed and immunolabelled with an antibody for pTrkB (red) and MAP2 (white). Nuclei were counterstained with Hoechst (blue). (**B and C**). Yellow arrows indicate BDNF-strep488 colocalizing with pTrkB in cell bodies (signaling endosomes). Big white arrowheads indicate signaling endosomes in dendrites (labeled by MAP2). The small white arrowhead in B indicated BDNF-strep488 vesicles containing no detectable levels of pTrkB Scale bar = 10 µm. Inset in B, four different regions of interest (ROI) are selected. ROI number 1 has BDNF-strep488 and detectable levels of pTrkB. ROI numbers 2 and 3 contain BDNF-strep488-positive endocytic structures with undetectable levels of pTrkB. ROI number 4 indicates a region considered background noise when quantifying pTrkB immunostaining. Scale bar = 1 µm. (**D and E**) Representative images of MAP2-positive dendrites containing signaling endosomes Scale bar = 10 µm. Inset, magnification of signaling endosomes were BDNF-strep488 (green) and pTrkB (red) are observed in close association with MAP2 immunostaining (white). Scale bar = 2µm. (**F**), the presence of detectable levels of pTrkB was quantified in 304 BDNF-strep488-positive endocytic structures, as indicated in the methodology section. BDNF-strep488 endosomes were classified by range size and the number of vesicles containing detectable levels of pTrkB was quantified (purple) in each range.

The online version of this article includes the following source data for figure 6:

**Source data 1.** Axonal BDNF colocalizes with active TrkB in signaling endosomes within cell bodies and dendrites of compartmentalized cortical neurons.

stimulation. We observed that BDNF increased the phosphorylation of 4E-BP1 at 90 min and that 4E-BP1 phosphorylation was even higher at 180 min and was down-regulated at 360 min (*Figure 8A and B*). Similar results were obtained for the phosphorylation of S6r (*Figure 8—figure supplement 1*). To evaluate whether activation of 4E-BP1 by axonal BDNF is dependent on the somatodendritic activity of PI3K and mTOR, we applied LY294002 or Torin1 to the CB of compartmentalized neurons (*Figure 8C*). The somatodendritic 4E-BP1 phosphorylation induced by axonal BDNF was significantly reduced by these inhibitors (*Figure 8D and E*). Additionally, treatment of axons with the dynein inhibitor Ciliobrevin D, reduced somatodendritic 4E-BP1 phosphorylation induced by axonal BDNF (*Figure 8D and E*). Together, these results indicate that upon arrival in the cell body, BDNF signaling endosomes induce activation of a signaling cascade that impacts mTOR-dependent protein synthesis.

The results presented above indicate that axonal BDNF induces the phosphorylation of CREB (*Figure 3*) and the activation of mTOR (*Figure 8*), suggesting that BDNF/TrkB signaling endosomes promote the translation of newly synthesized transcripts. To test this hypothesis, we metabolically labeled newly synthesized protein using Click-iT chemistry, which involves incorporation of L-azidohomoalanine (AHA). AHA is a modified amino acid that resembles methionine (*Dieterich et al., 2010*). In addition, we evaluated the level of de novo protein synthesis by using immunofluorescence to assess the levels of Arc, an immediate-early gene that encodes a protein required for synaptic plasticity induced by BDNF (*Ying et al., 2002*) and whose synthesis is regulated by CREB and mTOR (*Ying et al., 2002*; *Takei et al., 2004*). First, we adapted the Click-iT AHA method to noncompartmentalized neurons by stimulating the neurons with BDNF for 5 hr in the presence or absence of cycloheximide. As expected, BDNF increased AHA fluorescence, and cycloheximide significantly reduced the fluorescence of AHA (*Figure 9—figure supplement 1*). We removed methionine and B27 from the medium of compartmentalized neurons for 1 hr. Next, we added AHA to the entire chamber and BDNF exclusively to the AC for 5 hr and treated the CB with or without KG501 or Torin1 CB (*Figure 9A*). As shown in *Figure 9*, BDNF significantly increased both AHA incorporation into newly synthesized proteins and Arc levels in the cell body and dendrites (*Figure 9B–D*). Remarkably, inhibition of CREB and mTOR resulted in significant reductions in BDNF-induced AHA incorporation into newly synthesized proteins and Arc protein expression in the cell body (*Figure 9B–D*).

Together, our findings show that PI3K signaling is not required for the axonal transport of signaling endosomes along the axon of cortical neurons; but is needed to increase dendritic branching upon their arrival to cell bodies.

Also, our studies are the first to give evidence that axonal BDNF/TrkB signaling endosomes activate mTOR downstream effectors and increase protein translation upon reaching the cell body. This last activity depended on CREB and mTOR, indicating that axonal signaling endosomes control the transcription and translation in a coordinated fashion of a subset of proteins, including Arc, in cell bodies.

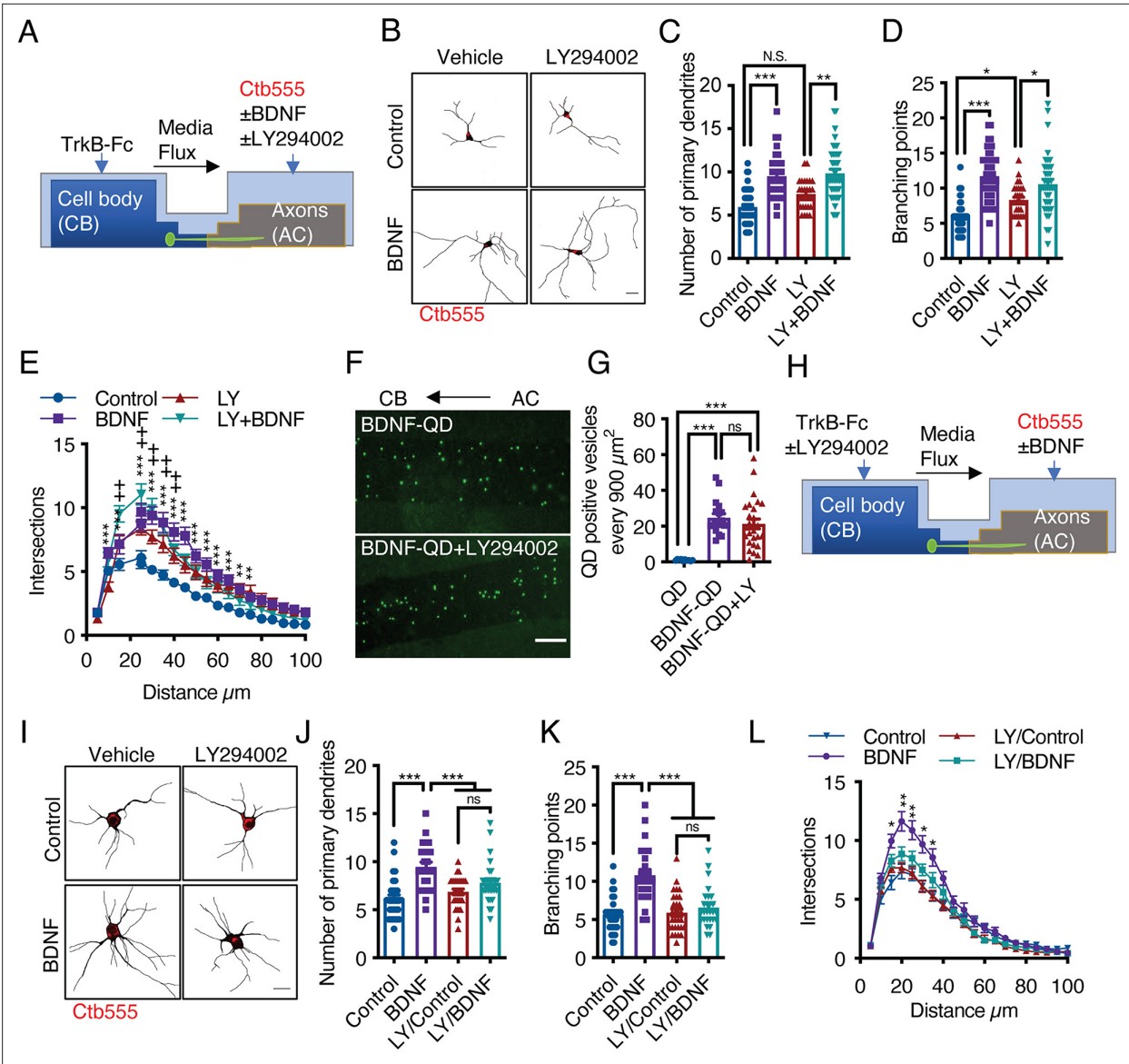

**Figure 7.** Activation of PI3K activity in the cell body but not the axons is required for dendritic arborization induced by axonal BDNF. (**A**) Schematic representation of the protocol used to stimulate compartmentalized cortical neurons. DIV six cortical neurons were transfected with EGFP, and then TrkB-Fc (100 ng/mL) was added to the CB and Ctb-555 and BDNF (50 ng/mL) with or without LY294002 (10 µM) were applied to the AC for 48 hr. Finally, the neurons were fixed, and immunofluorescence for MAP2 was performed. (**B**) Representative images of the CB (red indicates Ctb-555) of compartmentalized neurons whose axons were treated with DMSO in the presence or absence of BDNF and LY294002. Scale bar 20 µm. (**C–E**) Quantification of primary dendrites (**C**) and branching points (**D**) and Sholl analysis (**E**) for neurons labeled with EGFP/MAP2/Ctb-555 under the different experimental conditions described in B. n=29–48 neurons from three independent experiments. (**F**) BDNF-QDs were added to the AC for 4 hr in the presence or absence of LY294002 (10 µM) to promote the retrograde transport of BDNF. Then, the neurons were fixed, and the accumulation of BDNF-QDs in the proximal compartment of the microgrooves (MG) was evaluated. Representative image of accumulated BDNF-QDs (green dots) in the proximal part of MG under each treatment. Scale bar = 10 µm. (**G**) Quantification of QD-positive vesicles in every 900 µm² area of the proximal region of MG under each condition. Scale bar = 10 µm. n=36 MG from three independent experiments were analyzed. (**H**) Schematic representation of the protocol used to stimulate compartmentalized cortical neurons. DIV six cortical neurons were transfected with EGFP. Then, TrkB-Fc (100 ng/mL) was added to the CB in the presence or absence of LY294002 (10 µM), and Ctb-555 and BDNF (50 ng/mL) were applied to the AC for 48 hr. Finally, the neurons were fixed, and immunofluorescence for MAP2 was performed. (**I**) Representative images of the CB (red indicates Ctb-555) of compartmentalized neurons whose cell bodies were treated with DMSO (control) or LY294002 and whose axons were treated with or without BDNF. Scale bar = 20 µm. (**C–E**) Quantification of primary dendrites (**C**) and branching points (**D**) Sholl analysis (**E**) for neurons labeled with EGFP/MAP2/Ctb-555. n=25–30 neurons from three independent experiments. *p<0.05, **p<0.01, ***p<0.001,+p < 0.05 the LY294002 group vs. the LY294002/BDNF group. The data were analyzed by one-way ANOVA followed by Bonferroni's multiple comparisons post-hoc test (C, D, G, J, and K). Statistical analysis of

*Figure 7 continued on next page*

Figure 7 continued

the Sholl analysis data was performed by two-way ANOVA followed by Bonferroni's multiple comparisons post-hoc test. The results are expressed as the mean ± SEM.

The online version of this article includes the following source data and figure supplement(s) for figure 7:

**Source data 1.** Activation of PI3K activity in the cell body but not the axons is required for dendritic arborization induced by axonal BDNF.

**Figure supplement 1.** PI3K activation is required for BDNF signaling in cortical neurons.

**Figure supplement 1—source data 1.** PI3K activation is required for BDNF signaling in cortical neurons.

**Figure supplement 2.** PI3K and mTOR activity is required for activation of downstream TrkB signaling pathways.

**Figure supplement 2—source data 1.** This source data contains the original western blots that supports the *Figure 7—figure supplement 2*.

## Discussion

In this study, we demonstrated, using microfluidic chambers and compartmentalized cultures derived from different mouse mutants, that BDNF/TrkB signaling endosomes are transported along the axons of cortical neurons in a dynein- and TrkB-dependent manner to regulate dendritic arborization. Although long-distance communication between neurotrophins in the PNS has been well described (***Cosker and***

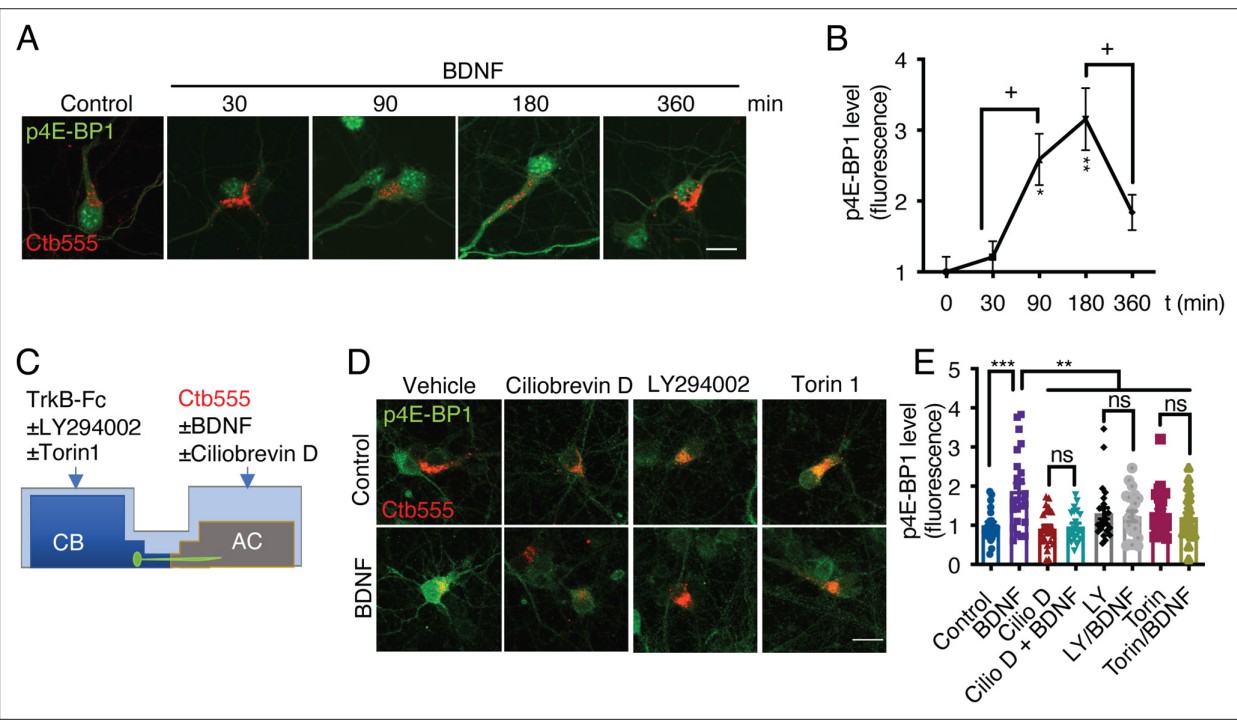

**Figure 8.** Axonal BDNF promotes mTOR activation in the cell bodies of compartmentalized cortical neurons in a PI3K-dependent manner. (**A**) Representative images of p4E-BP1 immunostaining (green) and Ctb-555 (red). DIV six compartmentalized cortical neurons were incubated with Ctb-555 overnight. At DIV 7, BDNF (50 ng/mL) was added to the AC of the neurons for 30, 60, 180, or 360 min. Scale bar = 20 μm. (**B**) Quantification of somatodendritic p4E-BP1 immunofluorescence in primary dendrites over time. (**C**) Schematic representation of the protocol used to evaluate the effect of different pharmacological inhibitors on the axonal BDNF-induced phosphorylation of 4E-BP1 in the cell body. DMSO (control), LY294002 (10 μm; LY), and Torin 1 (0.25 μm; Torin) were added to the CB of DIV six cortical neurons, or CilioD (20 μm; CilioD) was applied to the AC for 1 hr. Then, BDNF was added to the AC for 180 min in the presence or absence of these inhibitors. (**D**) Representative images of p4E-BP1 (green) in the somatodendritic compartment of neurons stimulated with BDNF in the presence or absence of different inhibitors. (**E**) Quantification of somatodendritic p4E-BP1 expression in neurons labeled with Ctb-555 (red) under each treatment. Scale bar = 20 μm. n=31–36 neurons from three independent experiments. *p<0.05, **p<0.01, ***p<0.001. vs. the control group;+p < 0.05 vs. the 90- and 360 min BDNF treatment groups in B. The data were analyzed by one-way ANOVA followed by Bonferroni's multiple comparisons post-hoc test. The results are expressed as the mean ± SEM.

The online version of this article includes the following source data and figure supplement(s) for figure 8:

**Source data 1.** Axonal BDNF promotes mTOR activation in the cell bodies of compartmentalized cortical neurons in a PI3K-dependent manner.

**Figure supplement 1.** BDNF added to axons increases pS6r in cell bodies.

**Figure supplement 1—source data 1.** BDNF added to axons increases pS6r in cell bodies.

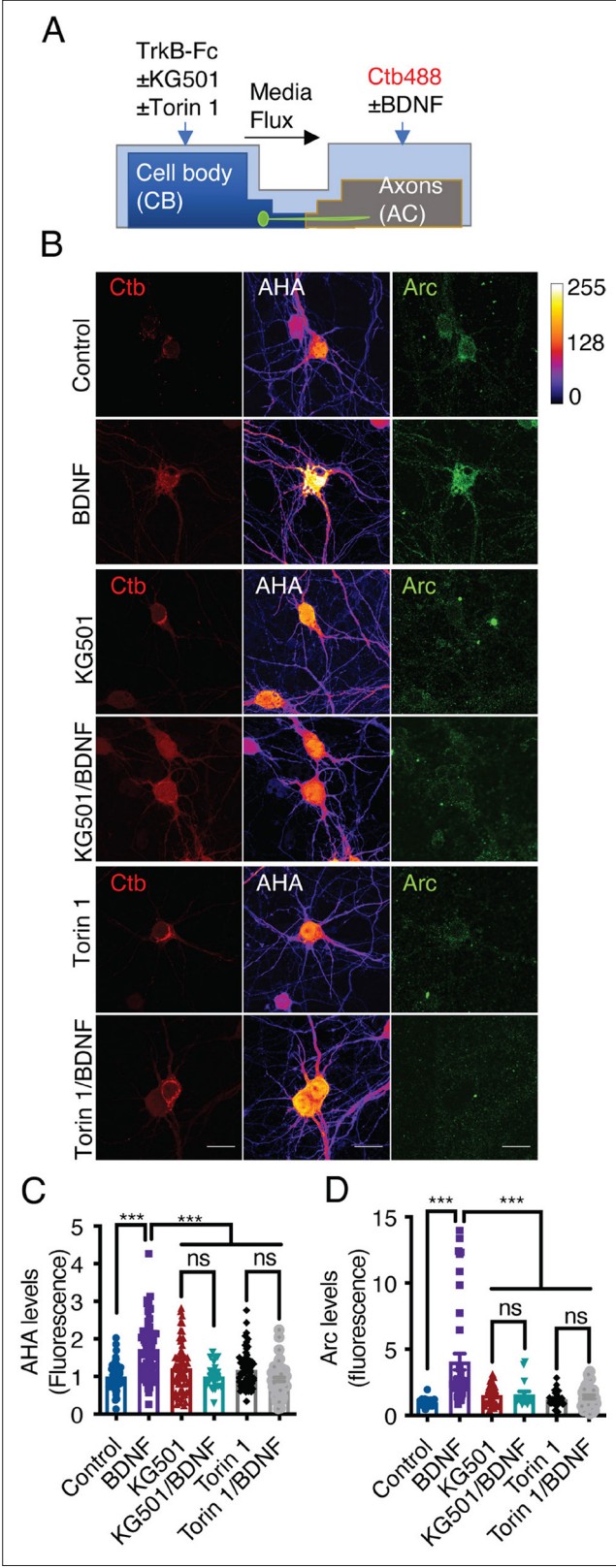

**Figure 9.** Axonal BDNF promotes somatodendritic protein synthesis in an mTOR- and CREB-dependent manner. (**A**) Schematic representation of the protocol used to evaluate protein synthesis. DIV 6–7 cortical neurons were incubated in methionine-free medium for 1 hr, and KG501 (10 µm) or Torin 1 (0.25 µm) was applied or not to the CB. Then, AHA was added to both the CB and AC, and BDNF (50 ng/mL) in the presence of Ctb-488 was added

*Figure 9 continued on next page*

*Figure 9 continued*

to only the AC for 5 hr. (**B**) Representative images of control and BDNF-stimulated neurons positive for Ctb-488 (red), AHA (in intensity color code mode, see color bar on the right) and Arc (green) in the presence or absence of KG501 and Torin 1. Scale bar = 20 μm (**C**) Quantification of AHA fluorescence in the primary dendrites of neurons labeled with Ctb-488 under each treatment. (**D**) Quantification of Arc fluorescence in the somas of neurons labeled with Ctb-488 under each different treatment. n=30–48 neurons from three independent experiments. ***p<0.001. The data were analyzed by one-way ANOVA followed by Bonferroni's multiple comparisons post-hoc test. The results are expressed as the mean ± SEM.

The online version of this article includes the following source data and figure supplement(s) for figure 9:

**Source data 1.** Axonal BDNF promotes somatodendritic protein synthesis in an mTOR- and CREB-dependent manner.

**Figure supplement 1.** Visualization of protein translation inusing AHA.

---

*Segal, 2014*; *Scott-Solomon and Kuruvilla, 2018*; *Harrington and Ginty, 2013*), whether signaling endosomes have a physiological role in the CNS in mammals, have not been addressed before. Our current study is the first to demonstrate direct evidence for BDNF long-distance signaling contributing to the shape of neuronal morphology in CNS neurons. We showed that this effect depends mainly on the activity of TrkB and not p75 in axons. Additionally, we positioned CREB activity as the central mediator of this effect. In PNS neurons, CREB activity has been associated mainly with neuronal survival and axonal growth induced by target-derived NGF (*Harrington and Ginty, 2013*) and in injured neurons to the increase in neuronal regeneration (*Caracciolo et al., 2018*). However, our study is the first to indicate that CREB activity, in the context of BDNF long-distance signaling, is mainly associated with BDNF-induced dendritic growth and not survival. Furthermore, our findings are the first to demonstrate that axonal BDNF and endogenous active TrkB are in signaling endosomes in cell bodies where their activity is required to propagate axonal BDNF signals to the nuclei of cortical neurons. Also, our work is the first report showing that BDNF/TrkB signaling endosomes generated in axons activate mTOR downstream effectors and increase general protein translation, including Arc, upon reaching the cell body. This last activity depended on CREB and mTOR, indicating that axonal signaling endosomes control the transcription and translation of a subset of proteins.

In this study, we have favored pharmacological inhibitors of the different components of downstream BDNF/TrkB signaling over genetic tools. The various drugs used allowed us to inhibit enzymatic or biological activities in a localized manner, that is, in the compartment of cell bodies or the axonal compartment of microfluidic cultures, since current genetic tools do not allow this type of evaluation. Complementary reagents were used to address the limitation in specificity that implies the use of drugs. For example, to evaluate axonal TrkB's participation in dendritic arborization induced by axonal BDNF we studied rat cultures using the Trks inhibitor K252A and in TrkBF616A knock-in mice cultures, the 1NM-PP1 kinase inhibitor. KG501 and a CREB dominant negative were used to assess CREB involvement. Finally, to evaluate the activation of p4EBP1 in cell bodies in the retrograde signaling of BDNF, the PI3K inhibitor LY294002 and mTOR inhibitor Torin 1 were used. In these experiments, the inhibitors abolished immunofluorescence, indicating a high specificity, at least for the evaluated substrate. Undoubtedly, pharmacological inhibitors have an explicit limitation of selectivity. There is still room for the development of inhibitors that can be used to assess local activities in neuronal cells, such as optogenetic inhibition, as it has been used for RabGTPases and the TrkA receptor (*Khamo et al., 2019*; *Nguyen et al., 2016*; *Cornejo et al., 2020*).

In this study, we clearly showed that BDNF/TrkB signaling endosomes originating in axons are required for dendritic arborization from the cell body. Our studies are in line with previous work showing that snapin, a protein that regulates autophagy and dynein transport, is required for dendritic branching in cortical neurons upon BDNF stimulation (*Cheng et al., 2015*; *Zheng et al., 2008*; *Zhou et al., 2012*) and with the fact that BDNF/TrkB signaling endosomes generated in presynaptic boutons in the hippocampus locally regulate neurotransmitter release (*Andres-Alonso et al., 2019*). Additionally, our results are related to the retrograde effects of BDNF on dendritic arborization observed in vivo in retinal ganglion cells in *Xenopus laevis* (*Lom et al., 2002*).

Using TrkB^F616A knock-in mice and 1NM-PP1, we showed that the application of BDNF to axons increases the transport of activated TrkB to the cell body and that activated somatic TrkB in neurons is required for nuclear responses. The fact that 1NM-PP1 reduced the activation of

already activated TrkB suggests that 1NM-PP1 binds the ATP-binding site of the receptor, reducing the tyrosine activity of TrkB without affecting its neurotrophin binding (*Chen et al., 2005*). There are two possible mechanisms that can explain our results. The first is that TrkB is phosphorylated and dephosphorylated during its transport along the axon; thus, 1NM-PP1 binds the ATP-binding cassette of the receptor and prevents its phosphorylation. Our finding of signaling endosomes containing axonal BDNF and active TrkB in cell bodies supports this possibility (*Figure 6*). Similar results were found in sympathetic neurons, where TrkA activated in distal axons is required to induce neuronal survival by axonal NGF (*Ye et al., 2003*). The second possible mechanism is that BDNF retrograde signaling promotes the phosphorylation of somatodendritic TrkB and that this process is required to maintain CREB phosphorylation. In agreement with this last idea, other groups have shown that retrogradely transported TrkA in sympathetic neurons locally accumulates in the somatodendritic compartment. The process increases the phosphorylation of other TrkA receptors in cell bodies (*Lehigh et al., 2017*; *Yamashita et al., 2017*). In addition, we also observed activated TrkB receptors positive vesicles not containing BDNF in axonally stimulated cortical neurons in our studies (unpublished observation). These results suggest that axonal BDNF/TrkB signaling endosomes are required to propagate BDNF signaling in cell bodies, which includes activating the cell body TrkB receptors.

Since BDNF can bind p75 and TrkB, we analyzed whether long-distance signaling is dependent on TrkB activation and the dependence of this process on p75 expression. p75 signaling has been reported to have an opposite role as Trk signaling since p75 can induce apoptosis or reduce axonal growth by binding to different ligands and receptors (*Ibáñez and Simi, 2012*; *Kraemer et al., 2014*; *Pathak et al., 2021*). Additionally, p75 negatively regulates dendritic complexity and spine density in hippocampal neurons (*Zagrebelsky et al., 2005*). We observed that the absence of p75 resulted in a decrease in basal dendritic arborization of cortical neurons in microfluidic devices. Nevertheless, BDNF was able to rescue this phenotype, suggesting that p75 participates in the regulation of dendritic arborization but is not required for axonal BDNF signaling.

We found that PI3K activity in axons does not play a role in the retrograde transport of signaling endosomes. It was speculated that PI3K plays a role in this process due to the role of phosphoinositide accumulation in early endosome-mediated regulation of the early-to-late transition process (transition of Rab5- to Rab7-positive endosomes; *Numrich and Ungermann, 2014*) required for retrograde sorting of signaling endosomes in motoneurons (*Deinhardt et al., 2006*). Our results are also different from findings regarding TrkA signaling in sympathetic neurons, as PI3K activity is required for the retrograde transport of TrkA signals (*Kuruvilla et al., 2000*). Nonetheless, the PI3K-mTOR signaling pathway is required for somatic responses to axonal BDNF. Indeed, our studies showed that PI3K activity is required for axonal BDNF-induced dendritic branching and phosphorylation of 4E-BP1 and S6r, which are downstream targets of mTOR, a process that is also dynein dependent. Consistent with these results, we demonstrated that axonal signaling endosomes activate the general translation of proteins in the cell body and that this process is dependent on CREB-mediated transcription and mTOR-dependent translation. These processes are accompanied by an increase in the protein levels of Arc. The fact that protein translation is also dependent on CREB suggests that CREB-dependent transcripts are specifically translated upon the arrival of signaling endosomes in the cell body in an mTOR-dependent fashion. *Arc* mRNA is anterogradely transported to dendrites to be locally synthesized (*Steward and Worley, 2002*), suggesting that axonal signaling may increase local translation in dendrites.

We performed a time-course study of CREB activation after application of BDNF to axons and found sustained activation of CREB after 3 hr of BDNF treatment. These results are different from what we have observed in noncompartmentalized hippocampal neurons, as CREB activity is decreased in these cells after 1 hr of BDNF treatment (*González-Gutiérrez et al., 2020*). While a dynein inhibitor reduced early activation of CREB by 40%, it reduced late CREB activation (after 3 hr of BDNF treatment) by 80%, suggesting that there is an additional component that contributes to the early activation of CREB by axonal BDNF in our study. In injured PNS neurons it has been observed calcium waves, as mediator of the nuclear regenerative response (*Cho et al., 2013*). It is possible that similar calcium waves are contributing to early CREB activation by axonal BDNF in our studies. Interestingly, it has been reported that both rapid and sustained CREB phosphorylation is required for proper transcriptional regulation (*Dolmetsch et al., 2001*), suggesting that in neurons, retrograde BDNF

signaling may contribute to sustained CREB phosphorylation for CREB-dependent maintenance of dendritic arborization.

CREB plays a central role in processes such as learning and memory (*Finkbeiner et al., 1997*; *Xing et al., 1998*; *Lonze and Ginty, 2002*). Indeed, mutations in several coregulators or downstream TFs are associated with genetic diseases leading to autism and cognitive disabilities (*Lyu et al., 2016*; *Zhou et al., 2016*; *McGirr et al., 2016*; *Wang et al., 2018*). Several studies have suggested that axonal BDNF can activate CREB in the cell body *Zhou et al., 2012*; *Watson et al., 1999*; *Deinhardt et al., 2006*; *Lazo et al., 2014*; however, no studies have addressed the physiological role of CREB activation induced by axonal BDNF signaling. In this work, we showed that CREB is required for long-distance BDNF-induced dendritic arborization in cortical neurons, suggesting that axonal BDNF signaling may contribute to the development and maintenance of different circuits in the brain, including those involved in learning and memory in the hippocampus and motor learning in the cortico-callosal and cortico-striatal pathways (*Andreska et al., 2020*; *Barco et al., 2005*; *Yap and Greenberg, 2018*).

Several studies have shown that CREB is required for the development of dendrites in vitro or during development (*Redmond et al., 2002*; *Kwon et al., 2011*; *Herzog et al., 2020*; *González-Gutiérrez et al., 2020*). In our studies, we showed that as observed in mice with conditional deletion of TrkB in cortical neurons (*Huang and Reichardt, 2003*), reduced CREB activity was required for the dendritic maintenance of pyramidal cells. This is consistent with a report indicating that CREB is required for cortical circuit plasticity after stroke (*Caracciolo et al., 2018*). However, the role of BDNF long-distance signaling regulating CREB activation in vivo remains to be demonstrated.

Together, our results demonstrate that BDNF/TrkB signaling endosomes in axons can coordinate CREB-dependent transcription and mTOR-dependent protein translation in the cell body, likely wiring circuits in the CNS.

## Materials and methods
### Materials
See *Supplementary file 1* for a list of all materials.

### Primary cortical neuron culture
Embryonic cortical neurons were obtained from mice (C57Bl/6 J) and rats (*Rattus norvegicus;* embryonic days 17–19) from the animal facilities of our institution. TrkB[F6161A] knock-in mice (Ntrk2 [tm1Ddg]/J strain#:022363), which have a single amino acid mutation in the intracellular ATP binding domain of TrkB that sensitizes the receptor to inhibition by 1NM-PP1 (*Chen et al., 2005*), and p75 KO[exonIII] heterozygous mice (B6.129S4-Ngfr[tm1Jae]/J strain#:002213) were purchased from The Jackson Laboratory (Sacramento, California, USA) and maintained as homozygotes (Ntrk2 [tm1Ddg]/J) or heterozygotes (B6.129S4-Ngfr[tm1Jae]/J). Pregnant animals were euthanized under deep anesthesia according to bioethical protocols approved by the Bioethics Committee of the Pontificia Universidad Catolica (protocol ID:180822013) de Chile and Universidad Andres Bello (act of approval 022/2019 and 009/2022).

Rat and mouse cortical tissues were dissected out and dissociated into single cells in HBSS. After disaggregation, the neurons were resuspended in modified Eagle's medium (MEM) supplemented with 10% horse serum (HS) (MEM/HS) and seeded in microfluidic chambers at a low density (40–50 x $10^3$ cells/chamber) or in mass culture at a density of $25 \times 10^3$ cells/well on 12 mm coverslips or $1.8 \times 10^6$ cells/60 mm plate. After 4 hr, the culture medium was replaced with neurobasal medium supplemented with 2% B27, 1 x GlutaMAX and 1 x penicillin/streptomycin. The proliferation of nonneuronal cells was limited by applying cytosine arabinoside (AraC; 0.25 µg/mL) when the MEM/HS was replaced with neurobasal medium and removed two days later (*Shimada et al., 1998*; *Taylor et al., 2003*).

### Microfluidic devices
The molds used to prepare the compartmentalized chambers were fabricated at the microfluidic core facility of Tel Aviv University (*Gluska et al., 2016*). The microfluidic chambers were prepared with a Syldgard 184 silicone elastomer base according to the manufacturer's instructions. Two days before primary culture, glass coverslips (25 mm) were incubated with poly-D-lysine (0.1 mg/mL). The next day, the poly-D-lysine was washed away, and microfluidic chambers with 400 µm microgrooves were

placed on the coverslips. Then, laminin (2 µg/mL in water) was added to the chamber. On the same day of primary culture, the laminin solution was replaced with Dulbecco's minimum essential medium (DMEM) supplemented with 10% HS, 1 x GlutaMAX and 1 x antibiotic/antimycotic (DMEM/HS).

## Quantification of BDNF-induced dendritic arborization in cortical neurons

For confocal microscopy analysis of cortical neurons in culture, a Nikon Eclipse C2 confocal microscope equipped with a digital camera connected to a computer with NIS-Elements C software was used. Images were acquired using a 60 x objective at a resolution of 1024x1024 pixels along the z-axis unless otherwise indicated. Cortical neurons (DIV 6) were transfected with 0.5 µg of a plasmid expressing EGFP using 0.8 µL of Lipofectamine 2000 in 30 µL of Opti-MEM. After 2 hr, the Opti-MEM containing the plasmid was replaced with neurobasal media supplemented with 2% B27, 1 x GlutaMAX, and 1 x penicillin/streptomycin for 1 hr. Neurobasal medium supplemented with TrkB-Fc (100 ng/mL) was applied to the CB for all treatments. 1NM-PP1 (1 µM), KG501 (10 µM), and LY294002 (10 µM) were added to the CB, and K252a (0.2 µM), LY294002 (10 µM), and CilioD (20 µM) were applied to the AC. After 1 hr, BDNF (50 ng/mL) and fluorescently labeled Ctb (1 µg/mL) were added to the AC. After 48 hr, the neurons were washed with PBS at 37 °C and then fixed with fresh 4% paraformaldehyde (PFA) in PBS (PFA-PBS) at 37 °C for 15 min. Then, the chamber was removed, and the neurons were permeabilized, blocked with 5% BSA and 0.5% Triton X-100 in PBS, and incubated with anti-MAP2 (1:500) in incubation solution (3% BSA and 0.1% Triton X-100 in PBS). After being washed 3 times, the neurons were incubated with Alexa 647-conjugated donkey anti-mouse (1:500) in incubation solution and mounted for analysis by fluorescence microscopy using Mowiol 4–88.

Dendritic arborization in cortical neurons labeled with Ctb, transfected with EGFP and labeled with MAP2 was analyzed. Primary dendrites and branching points were quantified, and Sholl analysis (*Sholl, 1953*) was performed (see below).

## Treatment of compartmentalized cortical cultures with adeno associate viral (AAV) particles

Mouse cortical neurons were cultured in microfluidic chambers using 50,000 cells/chamber in the presence of 0.250 µM Ara-C to arrest non-neuronal cell growth. At DIV 4, cells were cotransduced with adeno associate viral (AAV) serotype 1 (AAV1) particles driving the expression of EGFP, mCherry and a dominant negative mutant of CREB containing a EGFP tag (CREB-DN) (*González-Gutiérrez et al., 2020*). The expression of proteins was under the synapsin promotor. Cultures (cell bodies and axons) were treated with AAV1- EGFP/AAV1-mCherry or AAV1 CREB-DN /AAV1-mCherry diluted in fresh culture medium using 1:100 dilution for each virus. Then, TrkB-Fc (100 ng/mL) was added to the cell bodies of all chambers and axons were treated or not with BDNF (50 ng/mL) for the next four days. The treatment was reinforced at DIV 6. In addition, CTB-647 (1 µg/mL) was added to the axonal AC of DIV 7 cultures for the last 24 hr of treatment. Finally, at DIV 8, the cells were washed with PBS, fixed with 4% PFA and 4% Sucrose in PBS for 15 min at 37 °C. Coverslips were mounted onto Mowiol mounting medium containing Hoechst to stain nuclei.

## Determination of neurites density in compartmentalized cortical cultures expressing mCherry and EGFP or mCherry and CREB-DN

The neurites density in the CB compartment of neurons transduced with AAV1 as indicated above was analyzed with the Leica TCS SP8 confocal microscope, using the 40 x objective (with immersion oil) at a resolution of 1024x1024 pixels, obtaining 8–12 optical sections with a thickness of 1 µm in the "z" axis. Using ImageJ 2.9.0 software, z-stacks were integrated to measure maximum mCherry fluorescence intensity. The region used to calculate fluorescence intensity did not include cell bodies. Eight to ten chambers from three independent experiments per condition were analyzed by taking between 3 and 6 pictures per chamber. The mean intensity (fluorescence intensity/ROI area) was considered for each image. The average of all the pictures taken from each chamber was calculated to obtain the final value per chamber. Finally, the mean fluorescence intensity was standardized to one considering the average fluorescence intensity of the chamber expressing CREB-DN/mCherry, since it was the lowest value. Finally, the results were plotted with the GraphPad Prism 7 software, a two-way ANOVA and Turkey's multiple comparisons test was used for statistical analysis.

## Stereotaxic brain surgery in mice

Two-month-old male C57BL76J mice were deeply anesthetized using a mixture of ketamine/xylazine (150 mg/kg, 15 mg/kg) and placed in a stereotaxic frame (RWD, Life Science Co.). AAV1 expressing CREB-DN (*González-Gutiérrez et al., 2020*) or EGFP alone (control) and AAV1 expressing mCherry were coinjected unilaterally into II/III layer of the sensory-motor cortex at the following coordinates:+0.6 mm anteroposterior to bregma, –1.5 mm lateral bregma and –0.6 mm deep (*Watakabe et al., 2014*). The brains of control mice were coinjected with AAV1 -EGFP and AAV1-mCherry (0.5 μL of each and 1 μL of 0.9% NaCl), and the brains of experimental mice were coinjected with AAV1-CREB-DN and AAV1-mCherry (0.5 μL of each and 1 μL of 0.9% NaCl). $1x10^8$ PFU of each AAV1 was administered. To analyze dendritic arborization, the animals were sacrificed with a mixture of ketamine/xylazine (200 mg/kg, 20 mg/kg) in saline three weeks postinjection and perfused transcardially with 0.9% NaCl, followed by fixation with 4% PFA in phosphate buffer (pH 7.4). The mouse brains were dehydrated in 30% sucrose and sectioned at a thickness of 40 μm with a cryostat (Leica Microsystem), and the brain sections were immunostained for mCherry (1:1000).

## Analysis of dendritic arborization

Confocal images of cultured cells were acquired using a 60 x objective at a resolution of 1024x1024 pixels along the z-axis. A total of 5–7 optical sections over a width of 0.5 covering the whole neuron were obtained. Images of tissue sections were acquired with a confocal laser microscope (Leica TCS SP8) with a 40 x oil objective at a resolution of 1024x1024 pixels. Fifteen to 20 optical sections of isolated mCherry-labeled neurons in layer II/III of the sensory-motor cortex of injected mice were obtained over a width of 0.5 μm along the z-axis. Subsequently, the z-stacks were integrated, and neurons were manually drawn guided by the original fluorescence image using the pencil tool in ImageJ and segmented to obtain binary images. The numbers of total primary dendrites and branching points of all dendrites were manually counted from the segmented images using the ImageJ open source platform (*Schindelin et al., 2012*). For Sholl analysis, concentric circles with increasing diameters (10 μm per step for primary cultured cortical neurons and 5 μm for cortical neurons in tissue sections) were traced around the cell body, and the number of intersections between dendrites and circles was counted and plotted for each circle. The analysis was performed using Sholl analysis plug in ImageJ (*Ferreira et al., 2014*). In addition, for cortical neurons in tissue sections, the cell body size and apical dendrite diameter were measured from the segmented images using the straight tool in ImageJ. Additionally, the width and length of the cell body and apical diameter were measured 5, 25, and 50 μm from the start of the apical dendrite.

## Evaluation of protein phosphorylation by immunofluorescence

Cortical neurons (DIV 5–6) were incubated with Ctb-555 (Ctb, 1 μg/mL) overnight. For all treatments, neurobasal medium supplemented with TrkB-Fc (100 ng/mL) was added to the CB of DIV 6–7 cortical neurons. LY294002 (10 μM) and Torin 1 (0.25 μM) were added to the CB, and CilioD (20 μM) was applied to the AC. After 1 hr, BDNF (50 ng/mL) was added to the AC. After the time point indicated in the figure, the samples were fixed with 4% PFA and phosphatase inhibitor in PBS for 15 min. The samples were blocked and permeabilized in blocking solution (5% BSA, 0.3% Triton X-100, and 1 x phosphatase inhibitor in PBS) for 1 hr and incubated with antibodies overnight (4 °C) in incubation buffer (3% BSA and 0.1% Triton X-100 in PBS). The following antibodies were used: anti-pCREB (1:500), anti-pS6r (1:100), and anti-p4E-BP1 (1:500). The samples were incubated with Alexa 488-conjugated donkey anti-rabbit secondary antibody (1:500) for 90 min in BSA (3%) and Triton X-100 (0.1%) in PBS and then incubated with Hoechst 33342 (5 μg/mL) to visualize the nuclei. Entire neurons in the CB were visualized by confocal microscopy, and five to seven optical sections with a thickness of 0.5 μm thick from a single cell were analyzed.

## Evaluation of neuronal viability by TUNEL staining

For the detection of fragmented DNA, KG501 (10 μM) was applied to the CB and Ctb-555 (1 μg/mL) was added to the AC of DIV 6 cortical neurons cultured in neurobasal medium supplemented with B27 for 48 hr. Oligomycin A (10 μM) was added to the CB for 5 min as a positive control. The neurons were washed with PBS and fixed in 4% PFA in PBS for 15 min at room temperature. After that, coverslips to which neurons were attached were washed with PBS three times for 5 min each. The neurons were

permeabilized in ice-cold citrate/Triton solution (0.1% citrate and 0.1% Triton X100) for 2 min on ice. Then, the coverslips were washed twice to remove the permeabilization solution and incubated with TUNEL labeling mixture (In Situ Cell Death Detection Kit) for 1 hr at 37 °C. The samples were incubated with Hoechst (5 μg/mL). TUNEL staining was analyzed by confocal microscopy, and Hoechst-positive nuclei and TUNEL-positive apoptotic nuclei were counted by using the ImageJ platform.

## Evaluation of retrograde transport of TrkB from axons to cell bodies via immunoendocytosis of Flag-TrkB in microfluidic chambers

Cortical neurons (DIV 5) were transfected with 0.5 μg of plasmid expressing Flag-TrkB (gift from Prof. Francis Lee, NYU, USA) using 0.8 μL of Lipofectamine 2000 in 30 μL Opti-MEM per chamber. After 48 hr, cortical neurons (DIV 7) were incubated at 4 °C for 10 min, and then an anti-Flag antibody (1:750) was applied to the AC for 45 min at 4 °C. Then, neurons were briefly washed with warm neurobasal medium, and BDNF (50 ng/mL) was added to the AC for 180 min. Finally, the samples were fixed with 4% PFA and a phosphatase inhibitor in PBS for 15 min. The samples were blocked and permeabilized in blocking solution (5% BSA, 0.3% Triton X-100, and 1 x phosphatase inhibitor in PBS) for 1 hr and incubated with an anti-pTrkB (Y816) (1:200) or anti-Rab5 (1:250) antibody in incubation buffer (3% BSA and 0.1% Triton X-100 in PBS) overnight (4 °C). The samples were incubated with Alexa 488-conjugated donkey anti-rabbit (1:500) and Alexa 555-conjugated donkey anti-mouse (1:500) secondary antibodies for 90 min in incubation buffer. We visualized entire neurons in the CB and the microgroove compartment via confocal microscopy using a confocal microscope. Five to seven optical slices with a thickness of 0.5 μm were analyzed for each cell.

## Evaluation of somatodendritic TrkB activity in cortical neurons from TrkB^F616A mice in compartmentalized cultures

Ctb-555 (1 μg/mL) was applied to the AC of cortical neurons from TrkB^F616A mice (DIV 6) overnight. Then, the neurons were washed with neurobasal medium. For control and BDNF treatment, neurobasal medium (without B27) supplemented with TrkB-Fc (100 ng/mL) was added to the CB, and the AC was treated with neurobasal medium with and without BDNF (50 ng/mL) for 3 hr. To inhibit TrkB activity in the CB, the culture medium was removed from the CB and 1NM-PP1 (1 μM) was added to the CB for 1 hr; additional medium was added to the AC to avoid leakage of 1NM-PP1 into the AC. Next, the AC of the neurons was treated with or without BDNF (50 ng/mL), with additional medium being added to the AC, and 1NM-PP1 (1 μM) and TrkB-Fc (100 ng/mL) were applied to the CB for 3 hr. When the activity of TrkB in the AC was inhibited, 1NM-PP1 (1 μM) was added to the AC for 1 hr. Then, BDNF (50 ng/mL) was applied to the AC in the presence of 1NM-PP1 (in AC) and TrkB-Fc (in CB) for 3 hr, with additional medium being added to the CB. In all conditions, TrkB-Fc (100 ng/mL) in neurobasal medium without B27 was added to the CB. After 3 hr of BDNF treatment, the neurons were washed with 1 x PBS, and the cells were fixed with 4% PFA in PBS for 15 min at room temperature. Finally, immunofluorescence for phosphoproteins was performed as described above. The expression of pTrkB (Y816; 1:200) and pCREB (S133, 1:500) was evaluated. To assess pTrkB and pCREB levels, we visualized the CB of entire neurons by confocal microscopy. The number of pTrkB puncta in the cell body were quantified in Ctb-positive cells and normalized to the size of the cell body. pCREB levels were evaluated as described above.

## Evaluation of TrkB activity and downstream signaling in noncompartmentalized cortical neurons from TrkB^F616A mice

Cortical neurons from TrkB^F616A mice (DIV 7) were treated with or without 1NM-PP1 (1 μM) in neurobasal medium (without B27) for 1 hr. Then, control neurons were treated with vehicle for 1 hr; neurons in the treatment group were incubated with BDNF (50 ng/mL) for 30 min, rinsed twice with 1 x PBS and incubated with vehicle for 30 min; neurons in the 1NM-PP1 (Pre) group were incubated with BDNF in the presence of 1NM-PP1 for 1 hr; and neurons in the 1NM-PP1 (Post) group were stimulated with BDNF for 30 min. Next, the cells were rinsed twice with 1 x PBS and incubated with 1NM-PP1 for 30 min. Finally, TrkB and Akt activity was evaluated by Western blotting, and CREB, 4E-BP1 and S6r expression was evaluated by immunofluorescence. For western blotting, the cells were lysed with RIPA buffer (0.1% SDS, 0.5% NP40, 10 mM Tris-HCl (pH 7.5), 1 mM EDTA, 150 mM NaCl, and 0.5% deoxycholic acid) containing protease and phosphatase inhibitors. The cell extracts were subjected

to standard SDS gel electrophoresis and Western blotting protocols using anti-pTrkB (Y816, 1:1000), TrkB (1:1000), pAkt (1:1000), Akt (1:1000), and GAPDH (1:1000) antibodies. For immunofluorescence, the neurons were washed with 1 x PBS and fixed with 4% PFA in PBS for 15 min at room temperature. Finally, immunofluorescence for phosphorylated proteins was performed as described above using the following antibodies: anti-pCREB (S133; 1:500), anti-pS6r (S235/236, 1:100), anti-p4E-BP1 (S65, 1:500) and anti-β-III tubulin (1:750). We visualized the CB of neurons by confocal microscopy. Images were acquired using a 20 x objective for pCREB and a 60 x objective for 4E-BP1 and S6r at a resolution of 1024x1024 pixels along the z-axis of whole cells.

## Detection of signaling endosomes containing BDNF and active TrkB (pTrkB) in compartmentalized cortical neurons

Cortical neurons were cultured in microfluidic chambers at a density of 50,000 cells per chamber in the presence of 0.250 µM Ara-C to prevent the growth of non-neuronal cells. Commercially available biotinylated BDNF coupled to streptavidin DyLight488 (f-BDNF) was produced by incubating (37 °C) for 20 min, 6 µl of 6 µM solution of streptavidin DyLight488 and 2 µl of 6 µM solution of biotinylated BDNF diluted in 72 µl of neurobasal medium supplemented with 0,1% BSA (molar range 1:1). Then, this solution was diluted in the same buffer to a final concentration of approximately 6 nM (equivalent to approximately 150 ng/ml of BDNF) by adding 320 µl of the buffer. At DIV 8, TrkB-Fc (100 ng/mL) was added to the neuronal bodies for 60 min. Then, the axons were treated with f-BDNF for 6 hr. Subsequently, the cells were washed with PBS supplemented with protease/phosphatase inhibitors and fixed with 4% paraformaldehyde and 4% sucrose diluted in PBS for 15 min at 37 °C. Samples were blocked and permeabilized with a blocking solution (0.2% Saponin, 3% BSA, 5% fish gelatin dissolved in 1 X PBS) for 60 min. Cells were then incubated with rabbit anti pTrkB (Y674/675, 1:200 dilution) and mouse anti-Map2 (1:500 dilution) in the following buffer: saponin (0.02%), BSA (3%), fish gelatin (5%) dissolved in PBS overnight at 4 °C. Subsequently, the cells were incubated with donkey anti-Rabbit Alexa555 and donkey anti-mouse Alexa647 (1:500) in the same buffer. Samples were plated onto Mowiol mounting medium containing Hoescht.

The identification of f-BDNF and pTrkB was performed with a Leica TCS SP8 confocal microscope (UNAB microscopy facility), using the 63 x objective (with immersion oil) and 5 X digital zoom with a resolution of 1024x1024 pixels, obtaining 5–7 optical sections with a thickness of 1 µm in the "z" axis. For image analysis, the ImageJ software was used. Pictures from 6 different chamber from three independent experiments were taken, 304 f-BDNF-positive endosomes were considered for the study. All the f-BDNF positive vesicles were outlined, and the mean fluorescence intensity associated with the pTrkB label was measured (as indicated in *Figure 6B*). The pTrkB fluorescence intensity was measured in a similar area in other neuron regions using the MAP2 channel; these values were averaged and subtracted from the pTrkB fluorescence intensity (mean pTrkB minus mean background). The total number of vesicles with detectable levels of f-BDNF were divided into different size range, and the number of vesicles labeled for f-BDNF and pTrkB (>0) was calculated for each range (see *Figure 6F*).

## Evaluation of protein synthesis by Click-iT chemistry and Arc immunofluorescence

Ctb-488 was added to the AC of compartmentalized cortical neurons (6 DIV) overnight. At DIV 7, the neurons were washed with warm 1 x PBS and incubated with DMEM without methionine supplemented with GlutaMAX 1 x and L-cystine for 60 min in the presence or absence of Torin 1 (0.25 µM) or KG501 (10 µM). The medium was replaced with DMEM supplemented with Click-iT AHA (0.1 mM) for 5 hr. The AC of neurons was treated with BDNF (50 ng/mL), and the CB compartment was incubated with TrkB-Fc (100 ng/mL) and Torin 1 (0.25 µM) or KG501 (10 µM). The neurons were fixed with 4% PFA in PBS for 15 min. Then, alkyne-Alexa Fluor 555 (2.5 µM) was conjugated to AHA according to the manufacturer's instruction. Finally, immunofluorescence for Arc (1:300) was performed as described above.

## BDNF-avi production, purification, biotinylation, and Q-Dot conjugation

Production of BDNF-QDs was performed according to *Stuardo et al., 2020*. Briefly, HEK293FT cells were cultured in DMEM, 4.5 g/l glucose, 10% FBS, and 1% penicillin/streptomycin. For transfection, HEK293FT cells were grown in 20 plates of 15 cm until they reached 70% confluency. The medium was

replaced with 25 ml of high-glucose serum-free DMEM. For transfection, 30 μg of pcDNA3.1-BDNFavi (gift from Prof. Chengbiao Wu, UCSD, USA) was added to one ml of high-glucose DMEM, and then 90 μg of PEI (45 μL) was added. The mixture was incubated at room temperature for 25 min and added dropwise to the medium. Transfected HEK293FT cells were incubated at 37 °C in 5% $CO_2$ for 48 hr, and then the medium was collected for protein purification.

The medium was harvested and treated with 30 mM phosphate buffer (pH 8.0), 500 mM NaCl, 20 mM imidazole, and protease inhibitor cocktail. After incubation on ice for 15 min, the medium was cleared by centrifugation at 9500 rpm for 45 min using a Hettich 46 R centrifuge. Ni-NTA resin was rinsed twice with washing buffer (30 mM phosphate buffer (pH 8.0), 500 mM NaCl, 20 mM imidazole, and protease inhibitor cocktail) and added to the medium at a concentration of 0.3 ml Ni-NTA/100 ml media. Then, the samples were incubated for 180 min at 4 °C. The medium/Ni-NTA slurry was loaded onto a column, and the captured Ni-NTA resin was washed with 10 ml wash buffer and eluted with 1 ml elution buffer (30 mM phosphate buffer (pH 8.0), 500 mM NaCl, 300 mM imidazole, and protease inhibitors). The purity and concentration of BDNF-avi were assessed by Western blotting using a standard BDNF curve and an anti-BDNF antibody as done previously (*Stuardo et al., 2020*).

BDNF-avi was dialyzed against Sanderson water for 15 min. Then, BDNF-avi protein was biotinylated via an enzymatic reaction using the enzyme BirA. BDNF-avi (600–800 ng) was incubated with biotinylation buffer (50 mM bicine, 10 mM MgOAc, and 50 μM D-biotin) in the presence of 10 mM ATP and BirA-GST (BirA 1:1 BDNF-avi) for 1 hr at 30 °C with agitation at 600 rpm. Then, 10 mM ATP and BirA-GST were added again, and the samples were incubated for an additional hr at 30 °C with agitation at 600 rpm.

For coupling of BDNF-bt to streptavidin QDs and visualization in microfluidic devices, 1 μL of QD-655 was added to 10 μL of BDNF-bt (approx. 3 ng/μL) to allow the binding of BDNF-bt to streptavidin QD-655. Then, neurobasal medium was added to the BDNF-QDs to a final volume of 20 μL, and the mixture was incubated at room temperature in an orbital rocker for 30 min. To evaluate the role of PI3K and dynein in BDNF-QD transport along axons, maintenance medium was replaced with neurobasal medium without B27, and AC of neurons were treated for 1 hr in the presence or absence of LY294002 (10 μM) or Ciliobrevin D (20 μM), respectively. Then, BDNF-QDs (2 nM) were added to the AC in the presence or absence of LY294002 or CilioD for 4 hr. The neurons were fixed with 3% PFA in PBS for 15 min at room temperature, washed with PBS, and mounted in Mowiol. We visualized the proximal part of the microgrooves in the CB by confocal microscopy. Three to five optical sections (3-5) with a thickness of 0.25 μm were analyzed for each axon. Images were acquired using a 100 x objective at a resolution of 1024x1024 pixels along the z-axis. The number of BDNF-QDs or QDs alone was quantified in 90 μm$^2$ regions of axons located in the microgrooves.

## Statistical analysis

The results are expressed as the average ± standard error of the mean (SEM). Sholl analysis curves were analyzed by two-way repeated-measures ANOVA followed by Bonferroni's multiple comparisons test. Student's t test or one-way ANOVA followed by an appropriate multiple comparisons test was performed depending on the number of groups in the experiment. Details about the specific test used, level of significance, and number of replicates are indicated in each figure legend. Statistical analyses were performed using GraphPad Prism 7 (Scientific Software).

## Acknowledgements

The authors thank Carolina Ramirez for help with the compartmentalized cultures, Nicolas Stuardo for help with the BDNF monobiotinylation protocol and Carlos Ibañez for helpful reading of the manuscript. The authors gratefully acknowledge financial support from ANID (Agencia Nacional de Investigacion y Desarrollo) FONDECYT (grant N°1171137 and N°1221203 to FCB), the Basal Center of Excellence in Science and Technology (PIA BASAL AFB170005 and ACE210009), DGI-UNAB (DI-01-21/NUC) to FCB and a PhD fellowship from ANID to GMA. RT acknowledges FONDECYT postdoctoral grant N°3200800 from ANID and contract sectei/113/2022 from Sectei (Secretaria de Educacion, Ciencia, Tecnologia e Innovacion de la Ciudad de Mexico), Mexico.

# Additional information

## Funding

| Funder | Grant reference number | Author |
|---|---|---|
| Agencia Nacional de Investigación y Desarrollo | regular FONDECYT N°1221203 to FCB | Francisca C Bronfman |
| Agencia Nacional de Investigación y Desarrollo | Regular FONDECYT N°1171137 to FCB | Francisca C Bronfman |
| Agencia Nacional de Investigación y Desarrollo | PIA BASAL AFB170005 to FBC | Francisca C Bronfman |
| Agencia Nacional de Investigación y Desarrollo | PIA BASAL ACE210009 to FCB | Francisca C Bronfman |
| Agencia Nacional de Investigación y Desarrollo | PhD fellowship to GMA | Guillermo Moya-Alvarado |
| Agencia Nacional de Investigación y Desarrollo | FONDECYT postdoctoral grant N°3200800 to RTF | Reynaldo Tiburcio-Felix |
| Universidad Andrés Bello | DGI-UNAB (DI-01-21/NUC) to FCB | Francisca C Bronfman |
| Secretaría de Estado de Ciencia, Tecnología e Innovación | sectei/113/2022 to RTF | Reynaldo Tiburcio-Felix |

The funders had no role in study design, data collection and interpretation, or the decision to submit the work for publication.

## Author contributions

Guillermo Moya-Alvarado, Conceptualization, Investigation, Methodology, Writing – original draft, Writing – review and editing; Reynaldo Tiburcio-Felix, María Raquel Ibáñez, Alejandro A Aguirre-Soto, Investigation; Miguel V Guerra, Investigation, Methodology; Chengbiao Wu, Eran Perlson, Supervision, Methodology; William C Mobley, Conceptualization, Supervision; Francisca C Bronfman, Conceptualization, Data curation, Supervision, Funding acquisition, Writing – original draft, Project administration, Writing – review and editing

## Author ORCIDs

Guillermo Moya-Alvarado  http://orcid.org/0000-0002-1948-2156
Francisca C Bronfman  http://orcid.org/0000-0002-3716-8443

## Ethics

This study was performed in strict accordance with the recommendations in the Guide for the Care and Use of Laboratory Animals of the National Institutes of Health. All of the animals were handled according to approved institutional animal care in Pontificia Universidad Católica and Universidad Andres Bello. The protocol was approved by the Committee on the Ethics of Animal Experiments of the Pontificia Universidad Católica and Universidad Andres Bello. All surgery was performed under sodium pentobarbital anesthesia or ketamine/xylazine (150 mg/kg, 15 mg/kg) , and every effort was made to minimize suffering.

## Decision letter and Author response

Decision letter https://doi.org/10.7554/eLife.77455.sa1
Author response https://doi.org/10.7554/eLife.77455.sa2

# Additional files

## Supplementary files
- Transparent reporting form
- Supplementary file 1. List of materials and reagents.

## Data availability

All data generated or analyzed during the study are included in the manuscript as supporting file; Source Data have been provided for Figures 1 to Figure 9 and for Figure 3-figure supplement 1 to Figure 8-figure supplement 1.

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
