## [Editor Report]

The results in this study represent an advance over previous experiments since long distance signaling is demonstrated in brain neurons, whereas earlier work was concerned with peripheral neurons. Also, this article is the first to show in a conclusive manner that intracellular signaling in brain neurons results in increased transcription and protein production. The novelty is derived from long-distance signaling that is proposed for the development of circuits in the brain.

---

## [Decision Letter]

**Decision letter after peer review:**

Thank you for submitting your article "BDNF/TrkB signaling endosomes in axons coordinate CREB/mTOR activation and protein synthesis in the cell body to induce dendritic growth in cortical neurons" for consideration by *eLife*. Your article has been reviewed by 3 peer reviewers, and the evaluation has been overseen by a Reviewing Editor and Suzanne Pfeffer as the Senior Editor. The following individuals involved in the review of your submission have agreed to reveal their identity: Avraham Yaron (Reviewer #2); Wilma J Friedman (Reviewer #3).

The reviewers have discussed their reviews and major concerns with one another. The Reviewing Editor has drafted this summary of their discussions to help you prepare a revised submission.

Essential revisions:

Based upon the concerns raised by the reviewers, a number of issues need to be addressed in a revised manuscript-

1. Additional direct evidence for retrograde signaling of BDNF. BDNF applied to distal axons of cortical neurons induces dendrite growth in vitro. It is concluded that BDNF can signal retrogradely, not that it does. Additional data is needed to support this latter point.

2. Explain how this study is an advance over previous experiments in the PNS.

3. There is a dependence on the use of chemical inhibitors. Alternate methods and results to support the use of several of the inhibitors are needed. The limitations should be discussed.

4. The CREB data should be removed since it is downstream of many processes, including retrograde BDNF signaling. Also, there is no evidence for signaling endosomes in vivo.

The cell biology methodology is carried out in a solid fashion. Previous results were confined to peripheral neurons. In this case, cortical neurons are being analyzed. An effort should be made to emphasize the novelty of the studies. The requirement for target-derived neurotrophins has not been clearly established in the CNS.

*Reviewer #1 (Recommendations for the authors):*

1. One important limitation is that essentially all of these results have already been published in other studies examining BDNF-TrkB signaling, or especially in NGF-TrkA signaling in peripheral neurons. Therefore, this manuscript represents a more incremental study of more limited interest to developmental neurobiologists, and as such is more appropriate for a specialized neuroscience journal rather than *eLife*.

2. The authors do not present data demonstrating a role for BDNF-TrkB retrogradely-transported signaling endosomes in dendritic growth in vivo. The authors do present data in Figure 3 that CREB is required in vivo for dendrite growth in cortex, but this is not surprising given the importance of CREB in many growth and plastic events. While the paper doesn't absolutely need this data, and the compartmentalized culture experiments are well done, its absence limits the impact of this study.

3. In Figure 4, the 30-minute P-CREB signal was not completely blocked with Ciliobreven D, although it is at 180 minutes (Figure 4D-F), a point they allude to in the Discussion. This raises the question: does NM-PP1 inhibit P-CREB in the cell bodies at this early 30 minute time? Is there some rapid signaling cascade involved, such as calcium waves that they suggest in the Discussion?

4. In Figure S3, the P-TrkB westerns are not very convincing (the induction by BDNF). The quantifications look good, suggesting this Western blot is not representative of the data. Data showing a clear BDNF-induced TrkB phosphorylation should be included.

5. In the discussion, the authors debate whether TrkB kinase inhibition in cell bodies and dendrites (which blocks dendrite growth and signaling) is mechanistically due to continuous re-autophosphorylation of TrkB coming from axons, or TrkB from axons activating TrkB in cell bodies. The authors propose the latter and cite papers that support this. However, the former is clearly true in peripheral neurons with NGF-TrkA signaling (Ye et al., Cell, 2003). Both possibilities are plausible at this point and should be equally proposed.

6. The authors forgot to include the Cho et al. 2013 paper in the bibliography. Related to this paper, the authors cite it to propose that calcium waves may cause early CREB phosphorylation. This paper, however, showed retrogradely moving calcium waves after a peripheral nerve injury, not after neurotrophic factor signaling. This should be written more clearly in the discussion because it is misleading/confusing as currently written.

*Reviewer #2 (Recommendations for the authors):*

Figure 3A CREB is activated in 30 minutes, seems fast, is it in agreement with what is known about BDNF retrograde signaling?

Figure 3G, S2D – I don't understand how this in vivo data adds to the paper and I think the authors should take it out. Also, the authors write that there is nonapoptotic cell death (S2D) and then that CREB has no effect on neuronal survival.

As I noted in the public review there is heavy use of pharmacology, if the authors cannot back it up by other approaches they should mention the caveats.

The experiment in Figure 5 – really nice and clever use of the F616A mutant.

*Reviewer #3 (Recommendations for the authors):*

On line 161 they refer to cultures derived from p75 KOexonIII as p75+/+ and this is confusing.

The dose of 50 ng/ml of BDNF was used to treat the axons, was this an optimal dose determined from a dose-response analysis? In mass cultures, TrkB can be activated by 10 ng/ml BDNF, so it would be interesting to know whether this retrograde signaling required a higher dose.

In the Results section, they should clearly indicate the experiments that were performed in vivo, since all the other data are in vitro this was not immediately apparent. Moreover, in their presentation of the results from the in vivo experiment, they mention "nonapoptotic cell death, as indicated by fragmentation of the nucleus" and this phrase needs to be clarified since nuclear fragmentation is often indicative of apoptosis.

Line 83 – TrkB stands for tropomyosin receptor kinase, not tyrosine kinase receptor.

---

## [Author Response]

Essential revisions:1. Additional direct evidence for retrograde signaling of BDNF. BDNF applied to distal axons of cortical neurons induces dendrite growth in vitro. It is concluded that BDNF can signal retrogradely, not that it does. Additional data is needed to support this latter point.

We agree with the reviewers that this part of the paper needed to be stronger. We have performed additional experiments treating axons with fluorescently labeled BDNF and visualizing their accumulation in cell bodies and dendrites. Then, we mark these vesicles by immunostaining with an antibody against phosphorylated TrkB. This antibody is against tyrosines of the intracellular catalytic domain of TrkB (tyr674/675). The group of Martin Korte used the same antibody to follow pTrkB by immunostaining (https://doi.org/10.3389/fnmol.2022.945348https://doi.org/10.3389/fnmol.2022.945348). During the process of setting up these experiments, we tested several pTrkB antibodies and this one gave the stronger signal to noise ratio and allowed double labeling with BDNF. We have analyzed more than 300 BDNF-positive vesicles and found that more than 60 % contain detectable levels of pTrkB (new Figure 6).

Figure 5A and B show representative compartmentalized neurons transfected with Flag-TrkB and co-localization of TrkB receptors retrogradely transported (labeled with Flag antibodies in axons) with pTrkB (tyr816). The experiments explained in Figure 6 and Figures 5A and B, together with those in Figures 5 D-G, are solid evidence that signaling endosomes containing BDNF and its cognate active TrkB receptors trigger signaling cascades. Figure 5 D-G shows the results of experiments where axons or cell bodies of compartmentalized cultures, derived from TrkBF616A knock-in mice, were treated with the kinase inhibitor 1NM-PP1 and reduced activity of pTrkB and CREB was found in cell bodies treated with 1NM-PP1. Indicating that retrogradely transported BDNF/TrkB signaling endosomes triggers nuclear responses.

2. Explain how this study is an advance over previous experiments in the PNS.

We agree that in general the novelty of the work and the differences with PNS neurons required more emphasis. Therefore, we have added to the description of Results section, after the end of the different subsection, a short sentence that directly indicates the novelty. Also, we have overviewed the novelty of the findings and discusses the differences with PNS in the discussion.

We can summarize them in the following points.

a – Using pharmacology and genetics models, we demonstrated, for the first time, direct evidence for BDNF long-distance signaling contributing to dendritic arborization in neurons from the CNS. Also, we showed that this effect depends mainly on the activity of TrkB (and not p75) in axons. Most of the evidence supporting retrograde signaling in PNS neurons is related to neuronal survival, axonal growth, or dendritic arborization and is associated with TrkA signaling, not TrkB. One paper by the group of David Ginty (doi: 10.1016/j.celrep.2017.03.028) describes that endosomes containing active TrkA are found in dendrites in close association with synaptic markers. in vivo, inhibition of TrkA kinase activity in cell bodies decreased synaptic punctate in cervical ganglia. No studies directly demonstrated that adding NGF to axons increases dendrite lengths, dendritic branching, or increases the number of synapses.

b – We demonstrated for the first time that CREB activation is required for long-distance induction of dendritic branching by BDNF in compartmentalized cortical neurons in vitro*.* In PNS neurons, CREB is mainly related to preserving neuronal survival. We also showed that CREB was needed for sustaining normal neuronal dendritic arborization in long-projecting neurons, such as the pyramidal neurons of the sensory-motor cortex. These neurons require BDNF/TrkB signaling to support normal dendritic arbors. We agree that the role of BDNF long-distance signaling regulating CREB activation in vivo remains to be demonstrated and we have clearly stated this fact in the discussion.

c – Our findings are the first to demonstrate that axonal BDNF and endogenous active TrkB are in signaling endosomes in cell bodies where their activity is required to propagate axonal BDNF signals to the nuclei of CNS neurons.

d – Our findings are the first to show that PI3K signaling is not required for the axonal transport of signaling endosomes along the axon but is needed to increase dendritic branching upon their arrival to cell bodies. In PNS neurons, PI3K activity is required for the axonal transport of NGF signaling endosomes. Therefore, BDNF signaling endosomes may share signaling and transport pathways, but they may differ in the molecular pathways that regulate their traffic.

e – Our study is the first to give evidence that axonal BDNF/TrkB signaling endosomes activate mTOR downstream effectors and increase protein translation, including Arc protein, upon reaching the cell body. To our knowledge, no proof of mTOR activation and increase of protein synthesis (including Arc) are reported for PNS neurons.

f – Our study is the first to give evidence that the increase in protein translation mentioned above depends on CREB and mTOR activity, indicating that axonal signaling endosomes control the transcription and translation of a subset of proteins in a coordinated fashion, including Arc, in cell bodies. This has not been demonstrated for PNS neurons.

3. There is a dependence on the use of chemical inhibitors. Alternate methods and results to support the use of several of the inhibitors are needed. The limitations should be discussed.

We agree with the reviewers that, ideally, alternate methods are required to support the use of several inhibitors. For this reason, we have included experiments expressing CREB dominant negative mutant in the new Figure 3 that complement the findings with small molecule KG501, an inhibitor of CREB association with CBP. On the other hand, we have added a paragraph in the Discussion section explaining why we have used inhibitors and their limitation. This paragraph reads as follows:

“In this study, we have favored pharmacological inhibitors of the different components of downstream BDNF/TrkB signaling over genetic tools. The various drugs used allowed us to inhibit enzymatic or biological activities in a localized manner, that is, in the compartment of cell bodies or the axonal compartment of microfluidic cultures, since current genetic tools do not allow this type of evaluation. Complementary reagents were used to address the limitation in specificity that implies the use of drugs. For example, to evaluate axonal TrkB's participation in dendritic arborization induced by axonal BDNF we studied rat cultures using the Trks inhibitor K252A and in TrkBF616A knock-in mice cultures, the 1NM-PP1 kinase inhibitor. KG501 and a CREB dominant negative were used to assess CREB involvement. Finally, to evaluate the activation of p4EBP1 in cell bodies in the retrograde signaling of BDNF, the PI3K inhibitor LY294002 and mTOR inhibitor Torin 1 were used. In these experiments, the inhibitors abolished immunofluorescence, indicating a high specificity, at least for the evaluated substrate. Undoubtedly, pharmacological inhibitors have an explicit limitation of selectivity. There is still room for the development of inhibitors that can be used to assess local activities in neuronal cells, such as optogenetic inhibition, as it has been used for RabGTPases and the TrkA receptor (Khamo et al., 2019, Nguyen et al., 2016, Cornejo et al., 2020).”

4. The CREB data should be removed since it is downstream of many processes, including retrograde BDNF signaling. Also, there is no evidence for signaling endosomes in vivo.

We agree that the form in which these results were presented in the previous manuscript version gives space for confusion. We have prepared a new supplementary Figure (Figure 3—figure supplement 3) that gathers all the results performed in vivo by injecting AAV1 intracerebrally, driving the expression of a dominant negative mutant of CREB and mCherry (the same AAV used in new Figure 3G-J). These results complement the results obtained in new Figure 3G-J and they may be valuable information to other scientists. Also, we have stated in the discussion and the description of results that these results are not evidence for in vivo signaling endosomes.

Reviewer #1 (Recommendations for the authors):4. In Figure S3, the P-TrkB westerns are not very convincing (the induction by BDNF). The quantifications look good, suggesting this Western blot is not representative of the data. Data showing a clear BDNF-induced TrkB phosphorylation should be included.

We agree with the reviewer, and we have changed the Western blot in what is now new Figure 5—figure supplement 1.

5. In the discussion, the authors debate whether TrkB kinase inhibition in cell bodies and dendrites (which blocks dendrite growth and signaling) is mechanistically due to continuous re-autophosphorylation of TrkB coming from axons, or TrkB from axons activating TrkB in cell bodies. The authors propose the latter and cite papers that support this. However, the former is clearly true in peripheral neurons with NGF-TrkA signaling (Ye et al., Cell, 2003). Both possibilities are plausible at this point and should be equally proposed.

We agree with the reviewer, and we have now added a new paragraph in the discussion that include this paper and it reads as follow in the discussion of the new version of the article.

“The first is that TrkB is phosphorylated and dephosphorylated during its transport along the axon; thus, 1NM-PP1 binds the ATP binding cassette of the receptor and prevents its phosphorylation. Our finding of signaling endosomes containing axonal BDNF and active TrkB in cell bodies supports this possibility (Figure 6). Similar results were found in sympathetic neurons, where TrkA activated in distal axons is required to induce neuronal survival by axonal NGF (Ye et al., 2003). The second possible mechanism is that BDNF retrograde signaling promotes the phosphorylation of somatodendritic TrkB and that this process is required to maintain CREB phosphorylation. In agreement with this last idea, other groups have shown that retrogradely transported TrkA in sympathetic neurons locally accumulates in the somatodendritic compartment. The process increases the phosphorylation of other TrkA receptors in cell bodies (Lehigh et al., 2017, Yamashita et al., 2017). In addition, we also observed activated TrkB receptors positive vesicles not containing BDNF in axonally stimulated cortical neurons in our studies (unpublished observation). These results suggest that axonal BDNF/TrkB signaling endosomes are required to propagate BDNF signaling in cell bodies, which includes activating the cell body TrkB receptors”.

6. The authors forgot to include the Cho et al. 2013 paper in the bibliography. Related to this paper, the authors cite it to propose that calcium waves may cause early CREB phosphorylation. This paper, however, showed retrogradely moving calcium waves after a peripheral nerve injury, not after neurotrophic factor signaling. This should be written more clearly in the discussion because it is misleading/confusing as currently written.

We apologize for this mistake, and we have now included the paper in the bibliography and re-written the sentence as follow in the Discussion section of the manuscript.

“In injured PNS neurons it has been observed calcium waves, as mediator of the nuclear regenerative response (Cho et al., 2013). It is possible that similar calcium waves are contributing to early CREB activation by axonal BDNF in our studies”.

Reviewer #2 (Recommendations for the authors):Figure 3G, S2D – I don't understand how this in vivo data adds to the paper and I think the authors should take it out. Also, the authors write that there is nonapoptotic cell death (S2D) and then that CREB has no effect on neuronal survival.

See answer 4 to essential revisions. In addition, we have corrected the mentioned sentence and now read as follow.

“Apoptotic cell death, as indicated by fragmentation of the nucleus revealed by Hoechst staining, was not observed (Figure S4B), indicating that the expression of CREB-DN-EGFP did not induce apoptotic cell death, consistent with the in vitro results presented in Figure 3”.

As I noted in the public review there is heavy use of pharmacology, if the authors cannot back it up by other approaches they should mention the caveats.

See response number 3 to essential revisions.

Reviewer #3 (Recommendations for the authors):

We would like to mention that we have never observed pCREB immunostaining outside the nucleus. Still, it is possible that pCREB is present in signaling endosomes in undetectable levels.

On line 161 they refer to cultures derived from p75 KOexonIII as p75+/+ and this is confusing.

Thank to mention this, we have changed the sentence as follow.

“Using the same experimental design described in Figure 1A and B, we prepared cultures derived from the cross of p75 KO^exonIII^ heterozygous mice. The littermates were wild type (p75WT), heterozygous (p75HET) and homozygous (p75KO) mice”.

In the Results section, they should clearly indicate the experiments that were performed in vivo, since all the other data are in vitro this was not immediately apparent. Moreover, in their presentation of the results from the in vivo experiment, they mention "nonapoptotic cell death, as indicated by fragmentation of the nucleus" and this phrase needs to be clarified since nuclear fragmentation is often indicative of apoptosis.

See answer 4 to essential revisions and answer 3 to reviewer 2.

Line 83 – TrkB stands for tropomyosin receptor kinase, not tyrosine kinase receptor.

Thanks for mentioning this. We have changed it in the text.